# THUNDERKITTENS: SIMPLE, FAST, AND *Adorable* KERNELS

**Benjamin F. Spector**[1], **Simran Arora**[1], **Aaryan Singhal**[1], **Arjun Parthasarathy**[2],
**Daniel Y. Fu**[3,4], **Christopher Ré**[1]
[1] Stanford University, [2] Columbia University, [3] UCSD, [4] Together AI
{bfs, simarora, aaryan04,chrismre}@stanford.edu, arjun.p@columbia.edu, danfu@ucsd.edu

## ABSTRACT

The challenge of mapping AI architectures to GPU hardware is creating a critical bottleneck in AI progress. Despite substantial efforts, hand-written custom kernels fail to meet their theoretical performance thresholds, even on well-established operations like linear attention. The diverse capabilities of GPUs suggests we might we need a wide variety of techniques to achieve high performance. However, our work explores if a small number of key abstractions can drastically simplify the process. We present THUNDERKITTENS (TK), a framework for writing performant AI kernels while remaining easy to use. Our abstractions map to the three levels of the GPU hierarchy: (1) at the warp-level, we provide 16x16 matrix tiles as basic data structures and PyTorch-like operations, (2) at the thread-block level, we provide templates for asynchronously overlapping operations, and (3) at the grid-level, TK helps hide block launch, tear-down, and memory costs. We show the value of TK by providing simple & diverse kernels that match or outperform prior art. We match CuBLAS and FlashAttention-3 on GEMM and attention inference performance and outperform the strongest baselines by $10-40\%$ on attention backwards, $8\times$ on state space models, and $14\times$ on linear attention.

## 1 INTRODUCTION

AI is bottlenecked by the problem of efficiently mapping AI architectures onto accelerated GPU hardware. There has been a Cambrian explosion of ML architectures (Ho et al., 2020; Gu & Dao, 2023); however, the performance of these architectures remains substantially below their theoretical potential, despite substantial effort to develop *kernels*, or GPU implementations. Notably, kernel support has been poor even for softmax attention, which is used throughout industry. FlashAttention-2 (Dao, 2024) suffered a 47% performance degradation when translated to the H100 GPU, and it took over two years from the release of the H100 to develop FlashAttention-3 (Shah et al., 2024).

We are inspired by several approaches to supporting the development of AI kernels. Ideally, we would have a framework that supports high **performance** for a **breadth** of primitives, while being **easy to use**, learn from, and maintain. High performance C++ embedded libraries like NVIDIA CUTLASS/CuTe (NVIDIA, 2017) contain a myriad of nested templates, while compiler based approaches like Triton (Tillet et al., 2019) provide users with simpler interfaces, but fewer optimizations. We ask how broad and fast we can go by choosing a small and opinionated set of abstractions.

The main vector of growth for accelerated compute is in specialized matrix multiply units. On the NVIDIA H100 and NVIDIA A100 GPUs, BF16 tensor cores represent $15-16\times$ the FLOPs available relative to general-purpose BF16 / FP32 compute. Consequently, any high performance framework must prioritize keeping tensor cores at high utilization whenever possible. However, all kernels have non-tensor operations, too (like memory loads or the softmax in attention), and it is crucial to minimize their overhead. This proposition is at the heart of our approach.

To understand the complexities and opportunities in building a simple, yet high performance framework, we examine a simplified model of GPU parallelism, further detailed in section 2.[1]

1. **Warp-level parallelism:** Modern GPUs consist of tens of thousands of hardware threads which execute in parallel. Threads are organized into small groups, "warps", which execute instructions

---

[1]We discuss primarily NVIDIA, but the parallelism types hold across architectures, including AMD and Apple GPUs; we provide experiments on an Apple M2 Pro in Appendix B.

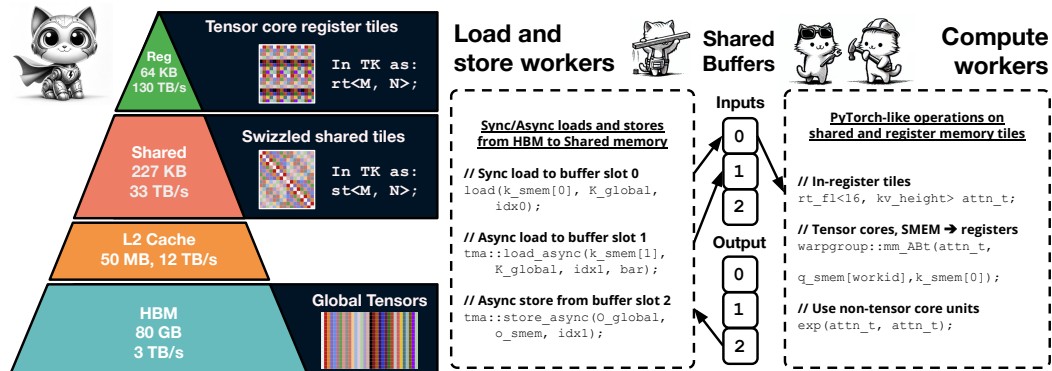

Figure 1: THUNDERKITTENS explores whether a small set of abstractions can enable performant AI kernels. Inspired by PyTorch, we first provide tiles with managed layouts and operations over these tiles. Second, we provide program templates for coordinating asynchronous workers – e.g., workers that load and store data, while other workers perform computations in fast memory.

together. Memory *layouts* determine how the logical data elements are mapped to physical thread ownership. If multiple threads try to access the same region ("bank") of memory, this can create expensive serializations between the threads (called "bank conflicts").

2. **Block-level parallelism:** Warps are grouped into "blocks" of threads, which can quickly share data. Warps execute their instructions on physical execution units, and having more warps in a block (called *occupancy*) can help run more instructions at the same time, reducing runtime. For example, one warp can run tensor cores for matmul, while another uses the ALU for $\max$.

3. **Grid-level parallelism.** GPUs run many blocks of threads at once, which communicate through large but slow global memory (HBM). An on-chip shared L2 cache helps reduce memory latencies and increase bandwidth if thread blocks reuse the same data. Thread blocks also face setup and tear-down overheads, which can introduce "pipeline bubbles" that hurt performance.

Despite the apparent need for a myriad of techniques to leverage all these hardware capabilities, our central technical finding is that indeed, for many AI kernels, *a small number of key abstractions exist that can simplify the process of writing high-performance kernels*. Our exploration led us to develop THUNDERKITTENS (TK), an AI kernel framework built around three key principles:

1. **Tile data structures with managed layouts:** Our interface is inspired by familiar ML frameworks like PyTorch and NumPy (Paszke et al., 2019), as highlighted in Figure 2. At the warp level, we use a $16 \times 16$ matrix tile as our basic data structure, maximizing compatibility with and encouraging the use of tensor cores. TK automatically picks the optimal memory layouts for the tiles to minimize bank conflicts while remaining compatible with specialized hardware instructions, avoiding user effort. We provide a set of parallel compute primitives over tiles, based on the suite of operations in PyTorch (e.g., pointwise $\text{multiply}$, $\text{mma}$, $\exp$, and $\text{cumsum}$ over tiles).

2. **Program template for asynchronous work:** At the block level, TK provides a general kernel template for coordinating asynchronous execution across warps in a thread block, built on the producer-consumer paradigm (Dijkstra, 1968). The developer's effort reduces to populating a few boilerplate functions within this model, using our PyTorch-like operands, and the template internally hides latencies through memory pipelines and synchronization primitives (Figure 1).

3. **Grid scheduling for pipelining thread-blocks.** At the grid level, we show TK can help developers reduce pipeline bubbles and improve L2 cache hit rates. Our template supports a *persistent grid*, where we overlap memory loads across thread block boundaries.

We highlight the value of these abstractions for developers in two ways:

- Through our exploration, we identify a few fundamental tradeoffs between achieving different types of parallelism including in setting the tile layouts (warp-level), occupancy (block level), and block launch order (grid level). Through our ablation studies (Section 3), we show how the simplified interface in TK gives users the control to navigate the tradeoffs.

- We validate the TK abstractions by providing kernels that match or outperform prior kernels for a range of AI operations. We match CuBLAS GEMMs and FlashAttention-3 attention inference,

PyTorch attention:

```python
1  # imports
2  import torch
3  import torch.nn.functional as F
4
5
6  # compute Q@K.T
7  att = torch.matmul(
8      q, k.transpose(2, 3))
9
10 # compute softmax
11 att = F.softmax(
12     att, dim=-1,
13     dtype=torch.float32)
14
15 # convert back to bf16
16 att = att.to(q.dtype)
17
18 # mma att@V
19 output = torch.matmul(att, v)
```

THUNDERKITTENS attention:

```cpp
1  // imports
2  using namespace kittens;
3  rt_bf<16, 64> k_reg, v_reg;
4  // load k from shared memory to register
5  load(k_reg, k_smem[subtile]);
6  // compute Q@K.T
7  zero(att);
8  mma_ABt(att, q_reg, k_reg, att);
9  // compute softmax
10 sub_row(att, att, max_vec);
11 exp(att, att);
12 div_row(att, att, norm_vec);
13 // convert to bf16 for mma_AB
14 copy(att_mma, att);
15 // load v from shared memory to register
16 load(v_reg, v_smem[subtile]);
17 auto &v_reg_c = swap_layout_inplace(v_reg);
18 // mma att@V onto o_reg
19 mma_AB(o_reg, att_mma, v_reg_c, o_reg);
```

Figure 2: A snippet of our attention kernel to show the PyTorch-like operations on tiles.

and outperform the strongest baselines by $10 - 40\%$ on attention backwards, up to $8\times$ on state space models, and up to $14\times$ on linear attention.

Our contributions are (1) showing a small and opinionated set of abstractions in TK that goes surprisingly far for writing simple and performant kernels; and (2) providing a collection of performant AI kernels. We hope that TK and its insights help improve the accessibility of AI kernels.

## 2  GPU FUNDAMENTALS

GPU tasks are divided into small programs called *kernels*. A kernel typically loads data from high bandwidth memory (HBM), performs work on it, and writes the outputs back to HBM before concluding. Before we explain THUNDERKITTENS's abstractions, we provide background on GPU parallelism at the *warp*, *block* and *grid* levels. We follow NVIDIA's terminology and focus on the H100 SXM GPU, though the principles apply across GPU vendors and generations.

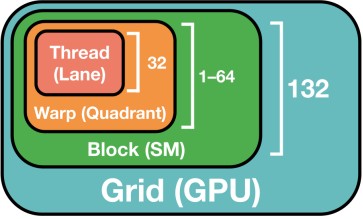

Figure 3: The software (and physical) GPU hierarchy.

### 2.1  GPU HIERARCHY

The GPU software hierarchy closely follows its physical hardware hierarchy (Figure 3). Here, we illustrate several of its most important components and aspects.

1. **Warps** consist of groups of 32 consecutive *threads* that operate on data in small but fast *register* memory. These instructions run on physical *execution units*, and individual threads can frequently occupy multiple specialized execution pipelines (below) at once through instruction-level parallelism, and different warps can further occupy available execution hardware:

   (a) Load and store units, to bring data into and out of registers. Advanced GPUs have also introduced dedicated hardware acceleration (e.g. H100 Tensor Memory Accelerator) for asynchronous bulk data movement between HBM and shared memory.

   (b) General purpose compute pipelines, such as **ALU** for $\max, \min$, **FMA** for multiplies and adds, and **XU** for complex operations like $\exp$. Throughput differs across the pipelines.

   (c) Accelerated matrix multiply hardware (tensor cores), which have most of the GPU compute.

   Threads can temporarily stall for a variety of reasons, including (but not limited to) fixed instruction latencies, memory latencies, barriers, pipeline throttles, or instruction cache misses.

2. **Thread blocks** are groups of warps which together execute a kernel on a physical core, called a *streaming multiprocessor* (SM). Although each SM has just four physical execution units, up to 64 software warps can simultaneously run on it (called "occupancy"). These collocated warps often contend on hardware resources: registers, shared memory, issue slots, and compute pipelines, but together they can help keep many work streams running at the same time within each execution unit. Warps *synchronize* at barriers, during which they cannot issue new work.

Importantly, warps within the same block can quickly communicate through special *shared memory* (SMEM, 227 KB, 33 TB/s). To improve bandwidth, SMEM is grouped into 32 physical "banks" of memory, which can serve memory simultaneously. However, if different threads try to access the same bank at the same time (called a *bank conflict*), their accesses must be serialized, which both increases access latencies and reduces available bandwidth. Hopper has limit of 255 registers per thread and attempts to request more, results in *spills* to the L1 cache. SMEM can be reallocated as an *L1 cache* for fast access to frequently used memory like spilled registers.

3. **Grids** of multiple thread blocks are launched to run the kernel. The H100 SXM GPU has 132 physical SM's which can run thread blocks at the same time. Although SM's are capable of collocating multiple thread blocks, most AI kernels can achieve high performance by simply collocating more warps within a single thread block (increasing the occupancy).

Thread blocks on the same GPU share common memory resources: large but slow high-bandwidth memory (80 GB, 3 TB/s), which has both the greatest latency and least bandwidth of all GPU memory, and a smaller but faster hardware-managed L2 cache (50 MB, 12 TB/s).

There are overheads to scheduling blocks. First, the block launch incurs *setup* costs and although this cost must be paid at least once at the initial kernel launch, kernels that continuously launch many large blocks can incur further costs. Second, there are *tail effect* costs if the grid is sized poorly. If a kernel of 133 blocks is executed on an H100 with 132 physical SMs, the kernel would require two waves to execute, the first with full efficiency, and the second with $< 1\%$ efficiency. More advanced schedules using independent CUDA streams can ameliorate these tail effects, such as recent work on asynchronous tensor-parallel schedules Chang et al. (2024).

## 2.2 COST MODEL

Summarizing the above components, we show a simplified cost model for GPU parallelism below. This cost model is inspired by the roofline model Williams et al. (2008). The overall kernel execution time $\mathbf{C}_{\text{Overall}}$ is the sum of the following costs where memory costs are a combination of the latency and bandwidth, and compute costs are a combination of latency and throughput.

$$\mathbf{C}_{\text{Overall}} = \max \Big( \underbrace{\mathbf{C}_{\text{HBM}}, \mathbf{C}_{\text{L2}}, \mathbf{C}_{\text{L1}}, \mathbf{C}_{\text{Shared}}}_{\textbf{Memory}}, \underbrace{\mathbf{C}_{\text{Tensor}}, \mathbf{C}_{\text{ALU}}, \mathbf{C}_{\text{FMA}}, \mathbf{C}_{\text{XU}}}_{\textbf{Compute}} \Big) + \underbrace{\mathbf{C}_{\text{Setup}} + \mathbf{C}_{\text{Sync}}}_{\textbf{Overhead}}$$

This model represents the *ideal case* of perfect overlapping between memory, compute, and tensor core costs. A kernel's actual cost will lie between the $\max$ and the $\text{sum}$ of the terms, depending on the workload properties (*i.e.*, some operations are inherently sequential), and the implementation efficiency. We aim to (1) reduce these individual costs, and (2) improve their collective overlapping. In Section 3, where we detail TK, we connect our primitives and optimizations back to these costs.

## 2.3 GPU PROGRAMMING FRAMEWORKS

We are inspired by a number of related efforts to simplify the development of AI kernels, such as NVIDIA CUTLASS/CuTe (NVIDIA, 2017) and Triton (Tillet et al., 2019).

CUTLASS's myriad of nested CUDA templates helps power highly optimized AI kernels (Shah et al., 2024; Bikshandi & Shah, 2023a;b) and fundamentally, the same kernels are expressible in TK and CUTLASS, since both are *embedded* libraries, giving users the full power of C++. We take a complementary approach by being rather opinionated about the abstractions. We ask: *(1) How far can we get with a small set of templates? and (2) Does concision sacrifice performance?* An appealing outcome is improved accessibility to AI researchers, since it can be challenging to fully leverage the capabilities of CUTLASS (Bikshandi & Shah, 2023b). We find that even industrially popular kernels written in CUTLASS, like FlashAttention-3, struggle from preventable issues like bank conflicts. We seek abstractions that manage such issues for users. Most recent AI architectures use high level compilers instead (Dao & Gu, 2024; Yang & Zhang, 2024; Fu et al., 2023c).

Triton, PyTorch (Paszke et al., 2019), TVM (Chen et al., 2018), TensorFlow XLA (Abadi et al., 2016), and others approach the problem from a compiler perspective. The frameworks are not C++ embedded, so it can be challenging to use unsupported specialized hardware instructions. It can also be difficult to manage asynchronous execution and register usage in high level frameworks. We explore avenues that retain the simple, PyTorch-like feel *while* enabling maximum performance in the next section. An extended discussion of related work is in Appendix A.

## 3 THUNDERKITTENS

We present THUNDERKITTENS (TK), a framework designed to simplify the development of high-performance AI kernels while leveraging the full capabilities of modern GPUs. This section (1) introduces our key programming abstractions and (2) shows how they can help developers navigate the tradeoffs between different types of parallelism. Section 3.1 focuses on warp level, Section 3.2 on thread block level, and Section 3.3 on grid level parallelism.

As running examples in this section, we show how TK helps optimize attention (Vaswani et al., 2017) and GEMM kernels. Section 4 demonstrates how the principles yield performant kernels for a breadth of AI operations (*e.g.*, attention variants, convolution, SSM, rotary).

### 3.1 WARP PARALLELISM WITH FAMILIAR DATA STRUCTURES AND OPERATIONS

At its core, THUNDERKITTENS is built on two fundamental abstractions – **tile data structures** at each level of the memory hierarchy and **bulk operands on tiles** akin to the familiar suite of operations in PyTorch and NumPy. We first define the abstractions, and then show they can help developers navigate tradeoffs between the tile *sizes* and compute efficiency.

**Programming abstractions** TK is heavily inspired by PyTorch and NumPy, given their familiarity to ML audiences (Paszke et al., 2019). We provide a concise set of parallel compute operations, based on the suite of operations in PyTorch (e.g., in Figure 2). The operations are executed by a "worker" abstraction, or a warp or warpgroup (4 warps) of threads that collaboratively own and operate on a piece of data. TK uses a $16 \times 16$ matrix tile as its basic data structure, designed to maximize compatibility with tensor cores. We provide tiles for each level of the memory hierarchy:

1. Register tiles and vectors, which are templated by type, shape, and layout. In Figure 2 we initialize a bfloat16 type tile with a column-major layout, height $16$, width $64$. The explicit control of register memory can help users reducing $\mathbf{C}_{\text{Memory}}$ in Section 2.
2. Shared tiles and vectors, which are templated by type and shape.
3. Global layout descriptors: We set up HBM loads and stores as indexing into $4D$ tensors, where the dimensions can be known at runtime or compile-time (saving valuable registers).

An advantage of these tile-based abstractions is that they enable TK to statically check layouts and operations, which is important because GPU kernels are often difficult to debug. For example, an in-register tensor core multiply mma_AB requires $A$ to be in a row-major layout, and $B$ to be in a column-major layout, and TK can raise compile-time errors if these conditions are not met.

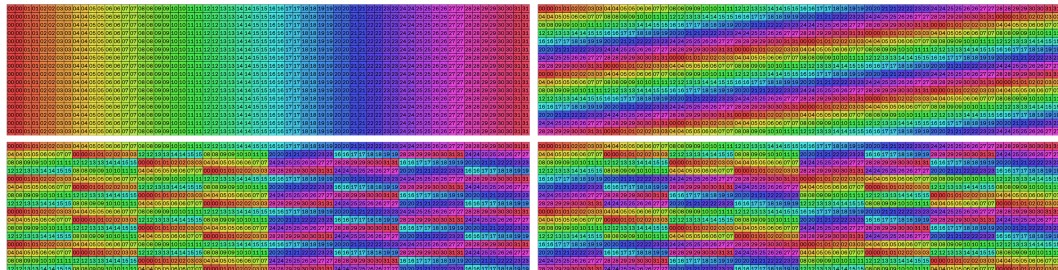

Figure 4: Shared memory bank layouts, illustrated for a 16x64 16-bit tile; each memory bank has its own color. **Top left:** A naive, row-major layout. Although loading rows is efficient, loading into a tensor core register layout suffers 8-way bank conflicts. **Top right:** A padded layout, which has no bank conflicts but consumes additional memory. **Bottom:** Two of TK's selected layouts, with compile-time selection based on width. (Bank conflicts are unavoidable for some tile sizes while maintaining good hardware support.) These layouts have 2-way and no bank conflicts, respectively.

**Choosing a memory layout** Layouts specify how logical data elements are mapped to physical thread ownership. Different tile sizes, types, and hardware-accelerated instructions benefit from different layouts, and some layouts lead to bank conflicts Our goals are:

- We want our register tiles (the fastest memory) to keep memory in the layouts used by tensor cores (the fastest compute). Shown in Figure 1 (Left); each color represents a different thread's ownership over the data. These formats are difficult to use, further highlighted in Figure 4.

- We want to support the use of hardware-accelerated instructions (*e.g.*, asynchronous matrix multiply and bulk copy instructions), which also require specific shared memory layouts.

In TK, we simplify to 3 layouts – swizzled on 32, 64, and 128 byte boundaries – and automatically assign shared tiles with layouts that minimize bank conflicts for their size and type. Seen in Section 4.2, even the FlashAttention-3 kernels written with CUTLASS templates can face bank conflicts, hurting performance. Our approach helps minimize conflicts, reducing $\mathbf{C}_{\text{Shared}}$ in Section 2.

### 3.2 BLOCK PARALLELISM WITH A GENERALIZED ASYNCHRONOUS TEMPLATE

THUNDERKITTENS helps developers reduce overheads by coordinating how workers in a thread block asynchronously overlap execution. Though the GPU hierarchy might suggest that we need a wide variety of techniques, we propose a *single* concise template that we find enables high performance on a surprisingly broad range of AI workloads. We first define the template, which has four steps – load-compute-store-finish (LCSF for short) – and builds on the classical producer-consumer paradigm (Dijkstra, 1968; Bauer et al., 2011). We show how the LCSF template can help navigate the tradeoffs between occupancy and efficiency (reducing $\mathbf{C}_{\text{HBM}}$, $\mathbf{C}_{\text{Compute}}$ in Section 2).

Load function:

```
1  if(warpgroup::warpid() == 0) {
2      tma::expect(inputs_arrived,
3          block.k, block.v);
4      tma::load_async(
5          block.k, globals.k,
6          {batch, head, iter, 0},
7          inputs_arrived);
8      tma::load_async(
9          block.v, globals.v,
10         {batch, head, iter, 0},
11         inputs_arrived);
12 }
13 else arrive(inputs_arrived);
```

Compute function:

```
1  warpgroup::mm_ABt(att, scratch.q[state.id], block.k);
2  warpgroup::mma_async_wait();
3
4  // softmax (simplified)
5  sub_row(att, att, max_vec);
6  exp(att, att);
7  div_row(att, att, norm_vec);
8
9  copy(att_bf16, att);
10
11 warpgroup::mma_AB(state.o, att_bf16, block.v);
12 warpgroup::mma_async_wait();
13 arrive(inputs_finished);
```

Figure 5: A simplified depiction of attention in the LCSF template to highlight the role of different specialized workers. Left is executed by workers that manage HBM to SRAM memory movement, and right by parallel compute workers, which operate in fast memory, registers and SRAM.

**Programming abstractions**   As per  Section 2, AI kernel usually load tiles of large tensors from HBM to SRAM, perform computation in fast memory, store the result for the tile back to HBM, and repeat this for the next tiles. To use the LCSF template, the developer writes four functions:

1. **Load function.** Specifies the data that load workers should load from HBM to shared memory, and when to signal to compute workers that this memory is ready for use.
2. **Compute function**. Specifies the kernel instructions that compute workers should execute, using the tile data structure and operation primitives from Section 3.1.
3. **Store function.** Specifies what data workers need to store to HBM.
4. **Finish function.** At the end of the kernel, the workers store any final state and exit.

TK provides abstractions to help the developer manage worker overlapping and synchronization.

1. Multi-stage buffer: The template maintains $N$-stage *pipelined buffers* in shared memory, which are used for loads and stores from HBM. Load/store workers add/remove tiles of data from the buffers, based on the status of compute workers. With a single stage, load workers would need to wait for all compute workers to finish executing before replacing the input tile. A 2-stage buffer can hide the HBM load (store) latency since the next tile can asynchronously load, while the compute work-

| M = N = K | Stages | TFLOPS |
|-----------|--------|--------|
| 4096      | 1      | 260    |
| 4096      | 2      | 484    |
| 4096      | 3      | 683    |
| 4096      | 4      | 760    |

Table 1: **Pipeline buffer stages** We measure efficiency in TFLOPS for our GEMM kernels as we vary the number of pipeline buffer stages in the TK template.

ers execute on the current tile. Deep buffers can reduce the synchronization required across compute workers, allowing them to operate on multiple tiles concurrently. TK lets the user set a single number to specify the number of stages, and manages the setup and use of these buffers for the user. In Section 3.2, we vary the number of stages $N \in \{1, 2, 3, 4\}$ for our GEMM kernel.

2. Synchronization barriers: Load/store workers need to alert compute workers when new memory is written to the input buffer. Compute workers need to alert load/store workers when tiles are written to the output buffer, or when input tiles can be evicted from the input buffer. Within the TK template, we provide an `arrive` function for workers to signal that they have finished their stage.

3. Asynchronous I/O: We wrap synchronous and asynchronous load and store instructions, including `cp.async` and TMA, in the same interface. We automate tensor map descriptor creation for TMA hardware-accelerated address generation for our global layout descriptors (`gl`).

**Tradeoffs between occupancy and efficiency**
TK parametrizes the *number* of load/store and compute workers (or occupancy) providing a simple way for developers tune their kernels. As discussed in Section 2, higher occupancy increases overlapping, but creates contention over limited hardware resources (e.g., registers). With fewer registers, workers need to operate on smaller tiles of data, resulting in more instruction issues, SRAM to register I/O, and potentially higher synchronization costs due to the increased data partitioning across workers.

Figure 6 shows the occupancy tradeoffs for attention kernels. We consider (1) a simple kernel that only uses warp level parallelism (Listing 2) and (2) a kernel written in the LCSF template (Listing 5). Although with both kernels, performance increases with occupancy until resource contention dominates, LCSF expands the Pareto frontier beyond the naive kernel.

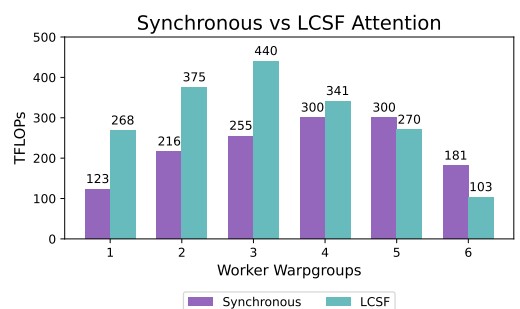

Figure 6: **Occupancy tradeoff: (Left)** Attention TFLOPs as a function of occupancy, benchmarked with head dimension 64 and context length 4096. We compare synchronous and LCSF kernels.

We find the general LCSF template to be effective across a range of AI workloads. We keep the template lightweight and simple by making opinionated design choices. However, we don't want TK to get in the way of achieving peak GPU performance – TK is *embedded*, meaning developers can use the full power of CUDA to extend the library as warranted.

### 3.3 GRID PARALLELISM WITH BLOCK LAUNCH SCHEDULING

TK makes it easier for users to quickly try varied grid layouts and coordinate thread block launches. This can help reduce the setup and tear-down costs for each thread block ($C_{Setup}$ in Section 2), and encourage memory reuse between thread blocks, to avoid slow HBM accesses ($C_{HBM}$ in Section 2).

**Block launch costs** We provide optimizations to minimize launch costs, centered around a *persistent grid*, where we launch thread blocks on the full set of SMs upfront, and simply load the next task for the kernel within the existing block. We further eliminate pipeline bubbles by having load/store workers anticipate the next task and pre-load memory to prepare for future work, while the compute workers run the finish stage for the prior task. Table 2 shows these optimizations for matrix multiplies.

| M=N | K | TK-No | TK-Yes | CuBLAS |
|------|------|-------|--------|--------|
| 4096 | 64 | 93 | 108 | 69 |
| 4096 | 128 | 161 | 184 | 133 |
| 4096 | 256 | 271 | 309 | 242 |
| 4096 | 512 | 414 | 450 | 407 |
| 4096 | 1024 | 565 | 600 | 633 |

Table 2: **Persistent block launch** TFLOPS for TK GEMM kernels with (**yes**) persistent and without (**no**) persistent launch as we vary matrix dimension $K$.

**L2 reuse and block launch order** Recall that thread blocks need to communicate via HBM. As introduced in Section 2, when thread blocks reuse memory, the data is often available in L2 cache, which is significantly faster than HBM. However, cache eviction means that these reuse qualities depend on the order in which blocks get launched. For our attention and GEMM kernels, we measure efficiency as we vary block order, summarized in Table 3. Block order substantially affects L2 reuse (measured through HBM bandwidth), which in turn can control kernel performance.

| Matrix Multiply (M=N=K=16384) | | | | Attention Forward (D=128) | | |
|---|---|---|---|---|---|---|
| Block Order | HBM GB/s | TFLOPS | | Block Order | HBM GB/s | TFLOPS |
| {8, N, M/8} | 982 | 805 | | {N, H, B} | 213 | 600 |
| {N, M} | 3,070 | 392 | | {B, H, N} | 2,390 | 494 |

Table 3: **L2 reuse:** We vary the block orders and measure both consumed bandwidth from HBM (GB/s) and efficiency (TFLOPS). For attention, we consider an optimized kernel, with an internal tiling of 8 rows of blocks, versus a naive kernel that schedules blocks in row-major order. For attention, we compare block order (1) sequence length $N$, heads $H$, and outermost batch $B$ vs. (2) innermost $B$, $H$, then outermost $N$. Block order has significant performance implications.

## 4 EXPERIMENTS

In experiments, we validate that THUNDERKITTENS speeds up a broad range of ML primitives. We compare to well-optimized kernels from prior work, written in alternate frameworks such as CUTLASS, CuBLAS, general CUDA, and Triton. We compare our kernels for the "workhorse" operations in AI, GEMM and attention, as well as kernels for emerging AI architectures, such as linear attention and state space models (Section 4.1). We profile the kernels to understand TK's role in achieving high performance in Section 4.2. Kernel listings, in the TK template, are in Appendix C.

### 4.1 TK ENABLES SIMPLE AND PERFORMANT AI KERNELS

We evaluate a suite of TK kernels. We benchmark on an NVIDIA H100 80GB SXM GPUs using CUDA 12.6 and report average TFLOPS. We provide experiments on an NVIDIA RTX 4090 and an Apple M2 Pro in Appendix B.

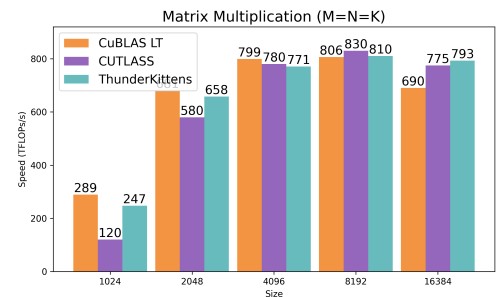

Figure 7: GEMM kernel from CuBLAS and TK.

**Workhorse kernels for AI** Industry teams and researchers have made significant investments into optimizing GEMMs and attention over the past several years (NVIDIA, 2023; Dao et al., 2022b; Bikshandi & Shah, 2023a; Shah et al., 2024, inter alia.), two workhorse operations that power the Transformer architecture (Vaswani et al., 2017). While the baselines are strong, TK kernels match or outperform:

- **GEMM:** We compare to the strongest baselines: CuBLAS(NVIDIA, 2023), CUTLASS NVIDIA (2017). A single TK matrix multiply kernel, with just 40 lines of device code, is competitive.
- **Attention:** We support multiple variants of attention: causal, non-causal, and grouped query attention (Ainslie et al., 2023) at head dimensions $64$ and $128$. We compare to the strongest baseline, which is concurrent to our work: FlashAttention-3 (FA3) (Shah et al., 2024). TK competes with FA3 across sequence lengths on the non-causal forwards pass, and outperforms FA3 on the causal and non-causal backwards pass by over $40\%$ at shorter and $10\%$ at longer sequences.

We find that TK makes it easy to use the GPU effectively by simplifying the choice of memory layouts, exploration of grid patterns for L2 reuse, and selection of occupancy and pipeline depth. The baseline kernels successfully use specialized H100 instructions and manage memory. However, the existing kernels are relatively complex: FlashAttention-3 proposes a "ping-pong scheduler" for workers, and the CuBLAS library is >600MB in CUDA 12.6 (Table 5), containing many tuned GEMM variants and logic to select the best option at runtime (Schuetze, 2024). With TK, we remove the ping-pong and maintain FA3-level efficiency, and we compete with CuBLAS on the demonstrated matrix sizes, using a single GEMM kernel (entirely in Appendix C.1).

**Kernels for emerging AI architectures** In addition to supporting peak performance on popular operations like GEMMs and attention, TK is also designed to be extensible to emerging AI workloads. We release kernels across recent ML primitives, including linear attention (Katharopoulos et al., 2020), FFT convolutions (Cooley & Tukey, 1965), and state space models (Gu et al., 2021).

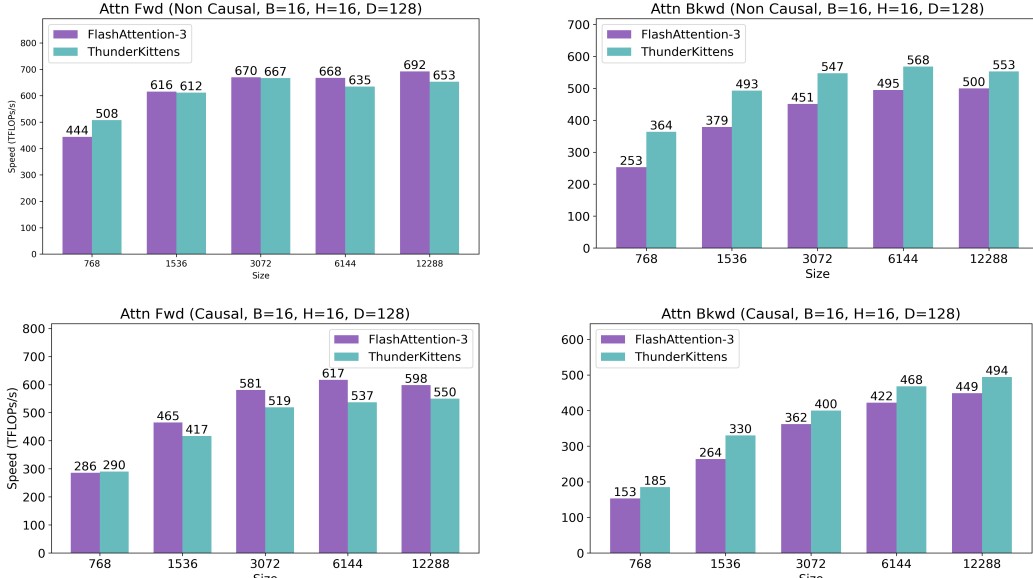

Figure 8: Attention causal and non causal inference and backwards pass efficiency.

- **Linear attention** We optimize two different classes of linear attention architectures, polynomial-based feature maps as in (Arora et al., 2024; Aksenov et al., 2024; Keles et al., 2023; Kacham et al., 2024) and *learned* feature maps as in (Zhang et al., 2024b;a). In Figure 9, we compare to the strongest available baselines: the popular Flash Linear Attention (FLA) CUDA kernels (Yang & Zhang, 2024), which are written in Triton. We show TK outperforms FLA's polynomial-based linear attention by $14\times$. TK outperforms FLA's learned map linear attention by $6.5\times$.

- **State space models** The long convolution, implemented with Fourier transforms using the convolution theorem, is the key primitive in popular state space modeling architectures such as S4, H3, and Hyena (Gu et al., 2021; Poli et al., 2023; Fu et al., 2023b; Hasani et al., 2022; Agarwal et al., 2024, inter alia.). In Figure 9, we compare to the strongest available baseline: the FlashFFTConv CUDA kernels in Fu et al. (2023c) and show TK outperforms the prior work by $4.7\times$ at sequence length $4096$ and $7.9\times$ at $1024$. TK outperforms PyTorch's FFT operations by up to $8.7\times$.

  We also optimize the recent Mamba-2 state space model (Dao & Gu, 2024). We provide a TK kernel that outperforms the Triton kernels in prior work Dao & Gu (2024) by $> 3\times$ (Figure 9). This gap is primarily due to the ease of fusing complex operations in TK.

We also develop kernels for common memory-intensive AI operations – fused dropout-residual-layernorm (Ba et al., 2016), and rotary (Su et al., 2023) – and show TK is effective. We compare to popular Triton kernels for these operations. [2]

**TK's programming model is extensible.** In Appendix B, we demonstrate that TK extends across (1) hardware platforms by providing competitive NVIDIA 4090 and Apple M2 kernels, (2) precisions by providing a competitive FP8 GEMM kernel, and (3) tile shapes by providing an attention kernel for arbitrary shapes.

## 4.2 COMPARING KERNEL IMPLEMENTATIONS

To understand TK's improvements, we profile the kernels using NVIDIA's NSight Compute (NCU) tool. In Table 4, we give NCU profiles for both the emerging long convolution primitive and the well-optimized attention backwards pass, comparing to the strongest respective baselines.

- **Long convolution:** We profile FlashFFTConv (FC) and TK convolution kernels at $B, D, N = 16, 1024, 4096$ in NCU. We find TK helps both with overlapping the workers (indicated by higher issue slots and fewer memory stalls) and in tensor core utilization ($4.1\times$ increase). This is enabled by our TK template, and use of TK warpgroup operations (which saves registers and establishes a SMEM to register memory pipeline through warpgroup matrix-multiply-add operations).

---

[2]Reference Triton kernels are from: `https://github.com/Dao-AILab/flash-attention`.

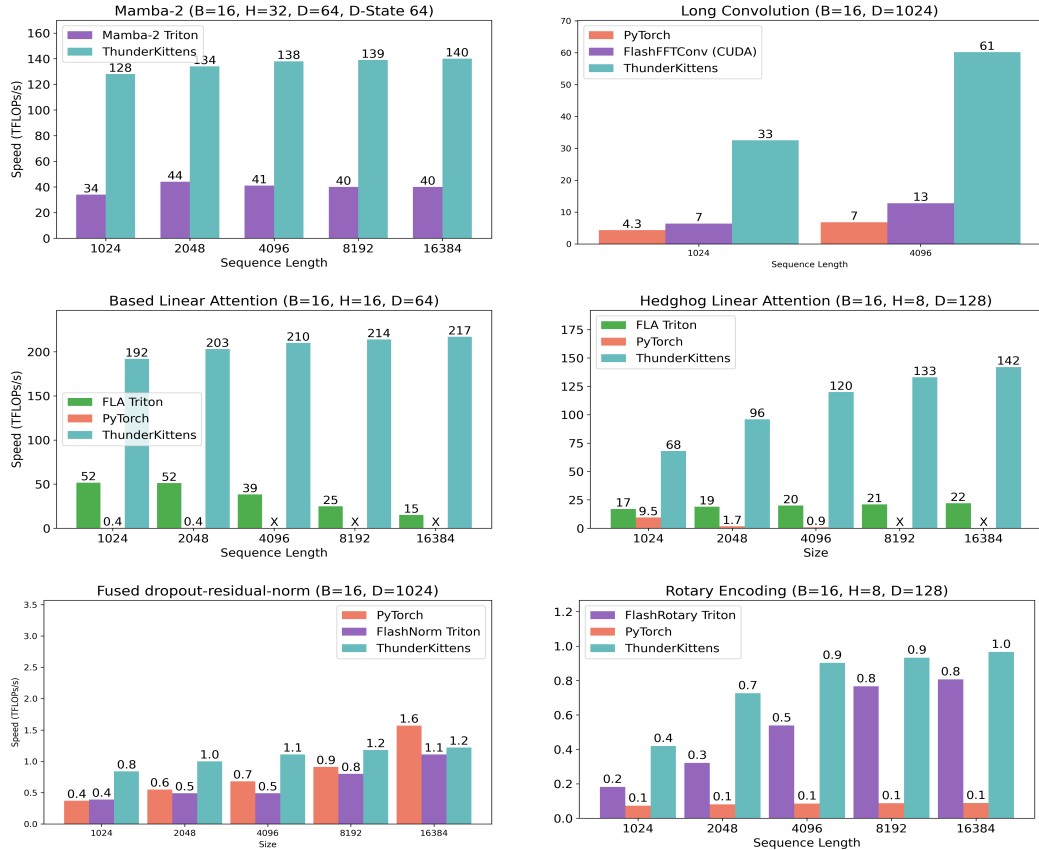

Figure 9: ThunderKittens kernels are performant across a wide range of kernels.

| Impl. | Occupancy utilizations (%) | | HBM | | Shared |
|---|---|---|---|---|---|
| | Tensor core | Issue slots | TPS (GB/s) | Stalls (Cycles) | Stalls (Cycles) |
| FA3 Bkwd | 61.2 | 25.1 | 328 | 1.83 | 0.92 |
| TK Bkwd | 58.2 | 34.8 | 490 | 1.63 | 0.14 |
| FlashFFT | 13.4 | 25.5 | 14.8 | 2.5 | 1.6 |
| TK | 54.8 | 40.0 | 31.4 | 0.6 | 0.3 |

Table 4: NCU profiles for 1) attention backwards pass kernels from FlashAttention-3 (Shah et al., 2024) vs. TK and 2) long convolution kernels from FlashFFTConv (Fu et al., 2023c) vs. TK.

- **Attention backwards:** We consider FA3 and TK at $B, H, N, D = 16, 16, 3072, 128$. The methods match in tensor core utilization, but TK gives higher issue slot utilization, suggesting the occupancy may be better-tuned. TK gives higher HBM memory throughput and incurs $10\%$ fewer stalled cycles on HBM waits. For shared memory, TK incurs $85\%$ fewer stalled cycles – we find TK has *no bank conflicts*, but NVIDA's NCU profiler reports up to 9.6-way bank conflicts in FA-3.

## 5 CONCLUSION

Given the challenge of mapping AI architectures to GPU hardware, our work asks how far we can get with a few, easy to use programming abstractions. In THUNDERKITTENS, we give abstractions for each level of the GPU hierarchy: tiles with managed layouts at the worker level, and an asynchronous LCSF template at the thread block level. We highlight tradeoffs for persistent block launches and L2 reuse at the grid level. The natural question is whether we sacrifice anything in performance when we write kernels with so few abstractions. We implement a breadth of AI kernels in TK and excitingly find that our abstractions are both general and consistently meet or exceed state-of-the-art. We are optimistic about the potential for accessible ways for programming AI hardware.

## 6 ACKNOWLEDGEMENTS

We are grateful to Together.ai for making this work possible. We than Arjun Parthasarthy for assistance in developing the complex tile support and FlashFFT kernels. We thank Mayee Chen, Tri Dao, Kawin Ethyarajh, Sabri Eyuboglu, Neel Guha, David Hou, Jordan Juravsky, Hermann Kumbong, Jerry Liu, Avner May, Quinn McIntyre, Jon Saad-Falcon, Vijay Thakkar, Albert Tseng, Michael Zhang for helpful feedback and discussions during this work. We gratefully acknowledge the support of NIH under No. U54EB020405 (Mobilize), NSF under Nos. CCF2247015 (Hardware-Aware), CCF1763315 (Beyond Sparsity), CCF1563078 (Volume to Velocity), and 1937301 (RTML); US DEVCOM ARL under Nos. W911NF-23-2-0184 (Long-context) and W911NF-21-2-0251 (Interactive Human-AI Teaming); ONR under Nos. N000142312633 (Deep Signal Processing); Stanford HAI under No. 247183; NXP, Xilinx, LETI-CEA, Intel, IBM, Microsoft, NEC, Toshiba, TSMC, ARM, Hitachi, BASF, Accenture, Ericsson, Qualcomm, Analog Devices, Google Cloud, Salesforce, Total, the HAI-GCP Cloud Credits for Research program, the Stanford Data Science Initiative (SDSI), BFS is supported by a Hertz Fellowship, SA is supported by a SGF Fellowship, and members of the Stanford DAWN project: Meta, Google, and VMWare. The U.S. Government is authorized to reproduce and distribute reprints for Governmental purposes notwithstanding any copyright notation thereon. Any opinions, findings, and conclusions or recommendations expressed in this material are those of the authors and do not necessarily reflect the views, policies, or endorsements, either expressed or implied, of NIH, ONR, or the U.S. Government.

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

Our appendix is organized as follows:

1. Appendix A provides an extended discussion of related work.
2. Appendix B provides a set of kernel that demonstrate the extensibility and simplicity of TK kernels.
3. Appendix C provides a set of kernel listings written in the TK abstractions.
4. Appendix D provides extended discussion of the TK library implementation details, including approaches to constructing shared memory layouts and the set of layouts used in TK.

## A   EXTENDED DISCUSSION OF PRIOR WORK

We provide an initial discussion of related work in Section 2 and extended discussion here. We discuss various programming frameworks that support AI kernel development and discuss prior work to develop hardware-aware AI algorithms.

**CPP embedded libraries**   Towards raising the level of abstraction and supporting simpler programs, NVIDIA maintains the CuTe and CUTLASS libraries, which are *CUDA* primitives library for graphics, scientific computing, and ML. As discussed in Section 2, both CUTLASS and TK can support the same kernels, since both are C++ embedded libraries. Developers can use the power of the full C++ library, including raw CUDA, when using these frameworks. Distinct from CUTLASS's objectives, we specifically explore how broad and fast we can go, just using a small number of opinionated abstractions.

**Compiler-based libraries**   Many machine learning frameworks employ high-level computational graph representations for optimizations, such as TVM Chen et al. (2018), TensorFlow XLA Abadi et al. (2016), Glow Rotem et al. (2018), and DLVM Wei et al. (2017). TVM, for instance, incorporates a flexible tensor expression language and automated schedule optimization. Building upon Halide's Ragan-Kelley et al. (2013) separation of algorithm and schedule, TVM introduces new primitives such as tensorization. The approaches of these frameworks differ from THUNDERKITTENS in that they provide a full end-to-end stack that optimizes both graph-level and operator-level transformations, while THUNDERKITTENS concentrates specifically on kernel-level optimizations.

Triton Tillet et al. (2019) builds on existing approaches to deep learning compilation while introducing novel techniques for efficient tiled computations on GPUs. Recently, tools such as Flex Attention also provide easy to use interfaces to write kernels for attention variants, and compile down to Triton He et al. (2024). Unlike XLA and Glow, which use tensor-level IRs and predefined templates, or Tensor Comprehensions Vasilache et al. (2018) and Diesel Elango et al. (2018), which rely on polyhedral models, Triton introduces a C-like language (Triton-C) and an LLVM-based IR (Triton-IR) centered around parametric tile variables. This approach supports non-affine tensor indices that polyhedral models struggle with. Triton's JIT compiler (Triton-JIT) implements tile-level optimizations and an auto-tuner, often enabling high performance on par with hand-tuned libraries. A key difference between TK and Triton is that TK is embedded within CUDA, and as a result its abstractions fail gracefully. In contrast, Triton supports inline PTX only for element-wise operations on tensors.

**Hardware-aware AI architectures**   We are inspired by the success of several prior works that introduce systems-level innovations to improve ML efficiency such as FlashAttention (Dao et al., 2022b; Dao, 2024; Shah et al., 2024) and other optimized attentions (Bikshandi & Shah, 2023a), FlashFFTConv (Fu et al., 2023c), linear attention kernels Vyas et al. (2020); Yang & Zhang (2024); Peng et al. (2023); Dao & Gu (2024). Given the effort required to independently optimize each new architecture, we ask whether a small set of opinionated GPU programming abstractions can help researchers obtain kernels for a broad range of AI operations.

## B   EXTENDED RESULTS

In Section 4, we show that THUNDERKITTENS results in state of the art kernels across a broad range of AI workloads on NVIDIA H100 GPUs. This section provides extended details of our

experimental protocol, additional results to analyze the extensibility and simplicity of TK kernels, and additional results to further highlight the breadth of features.

## B.1 EXTENDED DETAILS OF EXPERIMENTAL PROTOCOL

- **Benchmarking kernels** In order to ensure fair performance comparisons between TK kernels and others, we run 10 warm-up iterations then use cudaEvents to measure total kernel execution time over 10 benchmarking iterations. Reported performance is the average of the 10 benchmarking iterations.
- **Benchmarking PyTorch** We measure baseline PyTorch algorithm-implementations with and without torch.compile - a new compile is run any time the input configurations (sequence length, batch size, etc.) change. We report the maximum TFLOPS with torch.compile across settings.
- **Tuning kernels** Baseline GEMM kernels are tuned via a grid-search over the default execution parameters exposed (if any) and through auto-tuning methods (exposed via CuBLASLt) - for baselines, the maximum performance achieved is reported. Furthermore, for Triton kernels, we run triton.autotune over the default parameter configurations provided in baselines we compare TK kernels to. In order to avoid impacting performance measurements, kernel tuning is done in separate iterations prior to warmup and benchmarking.

We use the following software versions for benchmarking: CUDA 12.6, Triton version 3.00, and PyTorch version 2.4.

## B.2 ANALYZING THE EXTENSIBILITY OF TK ACROSS WORKLOADS

We include additional kernels to demonstrate two additional TK features:

1. **FP8 precision:** We provide an FP8 GEMM kernel in TK and compare this to CuBLAS in Figure 10. The inputs and outputs are both in FP8 precision, and the accumulate is in FP32.
2. **Padded tiles** While TK uses $16 \times 16$ tiles by default, to encourage users to utilize tensor cores and coalesced loads, it is important to support un-aligned workloads. TK handles this by padding the tiles (discussed in Appendix D). We implement attention with non-aligned dimensions on the NVIDIA 4090 and H100 GPUs and find that the performance characteristics remain the same as our aligned kernels.

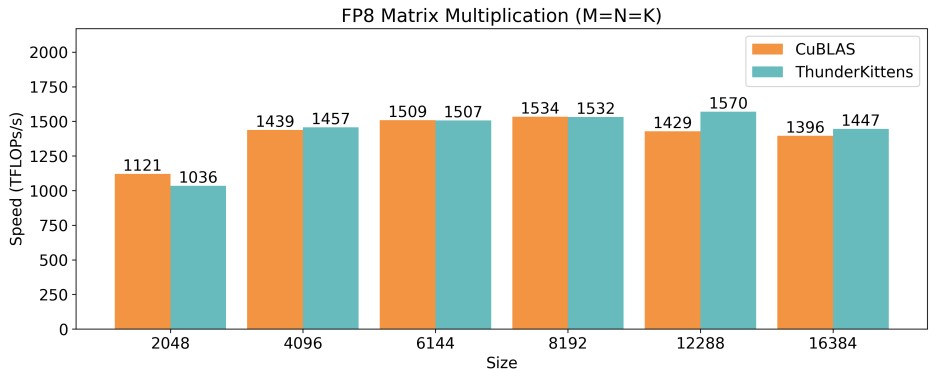

Figure 10: GEMM kernel using FP8 precision on the NVIDIA Hopper GPU.

## B.3 ANALYZING THE EXTENSIBILITY OF TK ACROSS HARDWARE PLATFORMS

We find that TK is extensible across hardware platforms. While we focused on the top-of-line *data center hardware*, NVIDIA H100, in Section 4, here we additionally consider the top-of-line *consumer hardware*, NVIDIA 4090, and *personal hardware*, Apple M2 chips.

**Consumer hardware: NVIDIA 4090 GPU** We implement non causal attention at head dimensions 64 and 128 using TK. We compare to FlashAttention-2 Dao (2024), a popular reference kernel, and find that the TK kernel is competitive across settings.

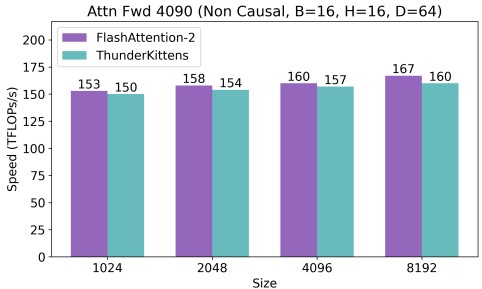
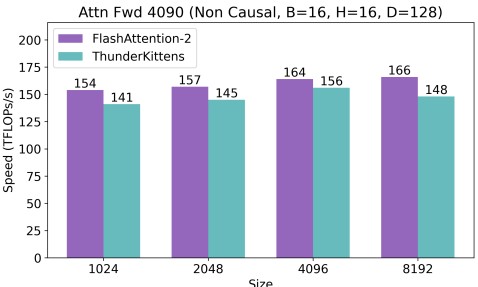

Figure 11: Attention non causal inference at head dimensions $64$ and $128$ and Based kernels, on NVIDIA 4090 chips using TK and the reference baselines.

**Personal hardware: Apple M2 chip** We implement non causal attention at head dimensions 64 and 128, and GEMMs on the M2 chip. We compare to the Apple MLX framework example kernels and find that the TK kernel is competitive across settings.

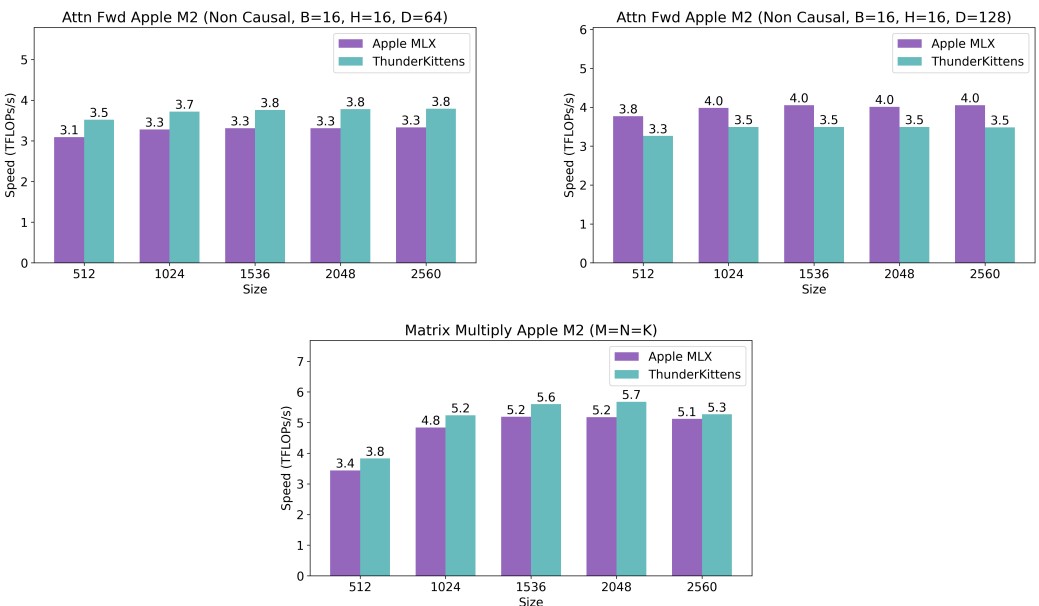

Figure 12: Attention non causal inference at head dimensions $64$ and $128$ and GEMM kernels, on Apple M2 chips using TK and the Apple MLX reference baselines.

We next include code listings for the attention kernel on NVIDIA 4090 fig. 13 and Apple M2 Figure 14, highlighting the close resemblance between the implementations.

NVIDIA 4090 attention:

```
1   template<int D> constexpr size_t ROWS = 16*(128/D); // height of each worker tile (rows)
2   template<int D, typename T=bf16, typename L=row_l> using qkvo_tile = rt<T, ROWS<D>, D, L>;
3   template<int D, typename T=float> using attn_tile = rt<T, ROWS<D>, ROWS<D>>;
4   template<int D> using shared_tile = st_bf<ROWS<D>, D>;
5   template<int D> using global_layout = gl<bf16, -1, -1, -1, D>; // B, N, H, at runtime, D at compile time
6   template<int D> struct globals { global_layout<D> Qg, Kg, Vg, Og; };
7
8   template<int D> __launch_bounds__(NUM_WORKERS*WARP_THREADS, 1)
9   __global__ void attend_ker(const __grid_constant__ globals<D> g) {
10
11      using load_group = kittens::group<2>; // pairs of workers collaboratively load k, v tiles
12      int loadid = load_group::groupid(), workerid = kittens::warpid(); // which worker am I?
13      constexpr int LOAD_BLOCKS = NUM_WORKERS / load_group::GROUP_WARPS;
14      const int batch = blockIdx.z, head = blockIdx.y, q_seq = blockIdx.x * NUM_WORKERS + workerid;
15
16      extern __shared__ alignment_dummy __shm[];
17      shared_allocator al((int*)&__shm[0]);
18      shared_tile<D>(&k_smem)[LOAD_BLOCKS][PIPE_STAGES] = al.allocate<shared_tile<D>,LOAD_BLOCKS,PIPE_STAGES>();
19      shared_tile<D>(&v_smem)[LOAD_BLOCKS][PIPE_STAGES] = al.allocate<shared_tile<D>,LOAD_BLOCKS,PIPE_STAGES>();
20      shared_tile<D> (&qo_smem)[NUM_WORKERS] = reinterpret_cast<shared_tile<D>(&)[NUM_WORKERS]>(k_smem);
21      // Initialize all of the register tiles.
22      qkvo_tile<D, bf16> q_reg, k_reg; // Q and K are both row layout, as we use mma_ABt
23      qkvo_tile<D, bf16, col_l> v_reg; // V is column layout, as we use mma_AB.
24      qkvo_tile<D, float> o_reg; // Output tile.
25      attn_tile<D, float> att_block; // attention tile, in float.
26      attn_tile<D, bf16> att_block_mma; // bf16 attention tile for second mma_AB. We cast right before.
27      typename attn_tile<D, float>::col_vec max_vec_last, max_vec, norm_vec; // these are column vectors.
28      // each warp loads its own Q tile of 16x64
29      if (q_seq*ROWS<D> < g.Qg.depth) {
30          load<1, false>(qo_smem[workerid], g.Qg, {batch, q_seq, head, 0}); // going through shared memory
31          __syncwarp();
32          load(q_reg, qo_smem[workerid]);
33      }
34      __syncthreads();
35
36      if constexpr(D == 64) mul(q_reg, q_reg, __float2bfloat16(0.125f * 1.44269504089));
37      else if constexpr(D == 128) mul(q_reg, q_reg, __float2bfloat16(0.08838834764f * 1.44269504089));
38      neg_infty(max_vec);
39      zero(norm_vec);
40      zero(o_reg);
41      // launch the load of the first k, v tiles
42      int kv_blocks = (g.Kg.depth + LOAD_BLOCKS*ROWS<D>-1) / (LOAD_BLOCKS*ROWS<D>), tic = 0;
43      load_group::load_async<1, false>(k_smem[loadid][0], g.Kg, {batch, loadid, head, 0});
44      load_group::load_async<1, false>(v_smem[loadid][0], g.Vg, {batch, loadid, head, 0});
45      // iterate over k, v for these q's that have been loaded
46      for(auto kv_idx = 0; kv_idx < kv_blocks; kv_idx++, tic=(tic+1)%3) {
47          int next_load_idx = (kv_idx+1)*LOAD_BLOCKS + loadid;
48          if(next_load_idx*ROWS<D> < g.Kg.depth) {
49              int next_tic = (tic+1)%3;
50              load_group::load_async<1, false>(k_smem[loadid][next_tic], g.Kg, {batch, next_load_idx, head, 0});
51              load_group::load_async<1, false>(v_smem[loadid][next_tic], g.Vg, {batch, next_load_idx, head, 0});
52              load_async_wait<1>(); // next k, v can stay in flight.
53          }
54          else load_async_wait();
55          __syncthreads();
56          #pragma unroll LOAD_BLOCKS
57          for(int subtile = 0; subtile < LOAD_BLOCKS && (kv_idx*LOAD_BLOCKS + subtile)*ROWS<D> < g.Kg.depth;
                 ↪ subtile++) {
58              load(k_reg, k_smem[subtile][tic]); // load k from shared into registers
59              zero(att_block); // zero 16x16 attention tile
60              mma_ABt(att_block, q_reg, k_reg, att_block); // Q@K.T
61              int first_index = (kv_idx*LOAD_BLOCKS + subtile)*ROWS<D>; // one past last KV index of tile
62              int start_fill = g.Kg.depth-first_index < ROWS<D> ? g.Kg.depth-first_index : ROWS<D>;
63              right_fill(att_block, att_block, start_fill, base_types::constants<float>::neg_infty());
64              copy(max_vec_last, max_vec);
65              row_max(max_vec, att_block, max_vec);
66              sub_row(att_block, att_block, max_vec);
67              exp2(att_block, att_block);
68              sub(max_vec_last, max_vec_last, max_vec);
69              exp2(max_vec_last, max_vec_last);
70              mul(norm_vec, norm_vec, max_vec_last);
71              row_sum(norm_vec, att_block, norm_vec);
72              copy(att_block_mma, att_block);
73              load(v_reg, v_smem[subtile][tic]);
74              mul_row(o_reg, o_reg, max_vec_last);
75              mma_AB(o_reg, att_block_mma, v_reg, o_reg);
76          }
77      }
78      div_row(o_reg, o_reg, norm_vec);
79      __syncthreads();
80      if (q_seq*ROWS<D> < g.Og.depth) { // write out o.
81          store(qo_smem[workerid], o_reg); // going through shared memory improves coalescing of dram writes.
82          __syncwarp();
83          store<1, false>(g.Og, qo_smem[workerid], {batch, q_seq, head, 0});
84      }
85  }
```

Figure 13: Attention implemented in TK on NVIDIA 4090 chips.

Apple M2 attention:

```
1   namespace custom_ops {
2   struct subexp2 {;
3       template<typename T> static METAL_FUNC T op(thread const T &a, thread        const T &b) { return metal::exp2(a-
            ↪ b); }
4   };
5   }
6
7   template<typename RT, typename RV>
8   static METAL_FUNC typename metal::enable_if<ducks::is_register_tile<RT>() && ducks::is_register_vector<RV>(), void
        ↪ >::type
9   subexp2(thread RT &dst, thread const RT &src, thread const RV &row_values) {
10      row_map<custom_ops::subexp2, RT, RV>(dst, src, row_values);
11  }
12  template<typename RV, typename U>
13  static METAL_FUNC typename metal::enable_if<ducks::is_register_vector<RV>(), void>::type
14  subexp2(thread RV &dst, thread const RV &lhs, thread const U &rhs) {
15      bin_op<custom_ops::subexp2, RV>(dst, lhs, rhs);
16  }
17  //constant constexpr const int D = 128;
18  #define NUM_WORKERS 1
19  template<int D>
20  kernel void attend_ker(ATTEND_KER_PARAMS) {
21      static_assert(D == 64 || D == 128, "D must be 64 or 128");
22      using global_layout = kittens::ore::gl<bfloat, 1, -1, -1, D>;
23      global_layout gl_q(__q__, nullptr, H, N, nullptr);
24      global_layout gl_k(__k__, nullptr, H, N, nullptr);
25      global_layout gl_v(__v__, nullptr, H, N, nullptr);
26      global_layout gl_o(__o__, nullptr, H, N, nullptr);
27      using st_qkv     = st_bf<8, D>;
28
29      using rt_qv     = rt_bf<8, D>;
30      using rt_k_t    = rt_bf<8, D, ducks::rt_layout::col>;
31      using rt_att    = rt_fl<8, 8>;
32      using rt_o      = rt_fl<8, D>;
33      using rv_att    = rt_fl<8, 8>::col_vec;
34
35      const int block = blockIdx.z;
36      const int head = blockIdx.y;
37      const int q_seq = (blockIdx.x * NUM_WORKERS) + warpId;
38      const int kv_blocks = N / st_qkv::rows;
39      rt_qv q_reg;
40      rt_k_t k_reg;
41      rt_qv v_reg;
42      rt_att att_block;
43      rt_o o_reg;
44      rv_att max_vec_last;
45      rv_att max_vec;
46      rv_att norm_vec;
47      load(q_reg, gl_q, {block, head, q_seq, 0}, laneId);
48      neg_infty(max_vec);
49      zero(norm_vec);
50      zero(o_reg);
51      constexpr const bf16 q_mul = ((D == 128) ? 0.08838834764bf : 0.125bf) * 1.44269504089bf;
52      mul(q_reg, q_reg, q_mul);
53      #pragma clang loop unroll(full)
54      for(auto kv_idx = 0; kv_idx < kv_blocks; kv_idx++) {
55              load(k_reg, gl_k, {block, head, kv_idx, 0}, laneId);
56              zero(att_block);
57              mma_ABt(att_block, q_reg, k_reg, att_block);
58              copy(max_vec_last,  max_vec, laneId);
59              row_max(max_vec, att_block, max_vec, laneId);
60              subexp2(max_vec_last, max_vec_last, max_vec);
61              subexp2(att_block, att_block, max_vec);
62              mul(norm_vec, norm_vec, max_vec_last);
63              row_sum(norm_vec, att_block, norm_vec, laneId);
64              mul_row(o_reg, o_reg, max_vec_last);
65              load(v_reg, gl_v, {block, head, kv_idx, 0}, laneId);
66              mma_AB(o_reg, att_block, v_reg, o_reg);
67      }
68      div_row(o_reg, o_reg, norm_vec);
69      store(gl_o, o_reg, {block, head, q_seq, 0}, laneId);
70  }
```

Figure 14: Attention implemented in TK on Apple M2 chips.

## B.4    ANALYZING THE SIMPLICITY OF TK

As a proxies for understanding the simplicity of the TK library, we measure (1) the size of various popular frameworks in bytes and (2) the lines of code across our kernels.

The library sizes are shown in Table 5. For CUTLASS and TK we report the size of the "include/" directory, and for Triton we report the combined size of the "include/" directories in Triton plus the "include/" in the core MLIR compiler dependency.

| Library | Size (Bytes) | Date / Version |
|---------|--------------|----------------|
| CUTLASS | 22 MB | 10/22/2024 |
| Triton | 12.6 MB | 10/22/2024 |
| TK | <1.0 MB | 10/22/2024 |

Table 5: Sizes of various CUDA libraries.

We find that the TK kernels in Table 6 average at $< 200$ lines of code. We compare to the lines of code in the corresponding state of the art baseline kernels, and the TK speed ups over these baselines. While measuring lines of code may be difficult, we provide links in the table indicate our approach. For TK, we include many comments, all the global data descriptor, and custom functions. We exclude the python bindings and other wrapper functions for all baselines. We generally observe that TK kernels use fewer lines of code and provide speed ups.

| Workload | TK kernel (LoC) | Reference kernel (LoC) | Speed up (min-max) |
|----------|-----------------|------------------------|--------------------|
| Attention forwards | 217 | 2325 (CUTLASS FA3) | 0.87-1.14$\times$ |
| GEMM | 84 | 463 (CUTLASS) | 0.98-2.05$\times$ |
| Convolution ($N = 4096$) | 131 | 624 (CUDA FlashFFTConv) | 4.7$\times$ |
| Based linear attention | 282 | 89 (Triton) | 3.7-14.5$\times$ |
| Hedgehog linear attention | 316 | 104 (Triton) | 4.0-6.5$\times$ |
| Mamba-2 | 192 | 532 (Triton) | 3.0-3.7$\times$ |
| Rotary | 101 | 119 (Triton) | 1.1-2.3$\times$ |
| Fused layernorm | 146 | 124 (Triton) | 1.0-2.2$\times$ |

Table 6: Lines of code (LoC) across TK H100 kernels, state of the art non TK kernels, and the TK speed up over the reference across the evaluated input dimensions in Section 4.

## C    THUNDERKITTENS KERNEL LISTINGS

This section first recaps our benchmarking methodology for the results and provides a set of kernels written in the TK LCSF template and tile abstractions:

1. Appendix C.1 GEMM kernel
2. Appendix C.2 Long convolution kernel
3. Appendix C.3 Attention kernel
4. Appendix C.4 Rotary kernel

To introduce the template components, we describe the GEMM kernel in detail in Appendix C.1.

**Benchmarking approach**    Our kernels in Section 4 are benchmarked on an NVIDIA H100 80GB SXM GPU with 10 warmup and 10 timed iterations using timings measured in C++. We also provide Python-bound kernels and benchmarking infrastructure in our repository for reference.

### C.1    MATRIX MULTIPLY

First we show and describe a TK GEMM kernel in the LCSF template.

Each compute warpgroup is responsible for computing $64M$-row, $64N$-column chunk of the resulting output matrix. Each compute worker identifies the coordinates for its chunk, zeros its accumulator registers, repeatedly runs large asynchronous matrix multiplies (compute), and finally stores out its tile in the end (finish). The load workers also compute their coordinates, and then repeatedly load chunks of the input matrices (load). Store workers perform asynchronous stores when the compute workers are finished with the chunks (stores).

**Tuning the number of workers and pipeline stages**    The computation is divided into stages, with each stage processing 64 elements along the reduction dimensions of the input matrices. The input pipeline is automatically sized by THUNDERKITTENS if the user does not specify a value. For common configurations of either a (2 compute warpgroup) $128 \times 256$ or (3 compute warpgroups) $192 \times 192$ output tile per block, it generates a 4-stage pipeline.

**Tuning the grid order**    The greatest complexity of this kernel is in setting the grid parameters. This kernel adopts a 3D stride over the input matrices, which has a significant effect for large matrices which do not fit in L2 cache. The order in which blocks execute strongly influences cache locality and thus available memory bandwidth. To illustrate the magnitude of the effect, comparing the presented scheme versus a naive grid (in which blocks are executed in row-major order) a $4096 \times 4096 \times 4096$ matrix multiply only drops from 767 TFLOPs to 735 TFLOPs, but a $16384 \times 16384 \times 16384$ matrix multiply drops from 797 TFLOPs to 387 TFLOPs, a $> 50\%$ performance degradation.

```
1   using namespace kittens;
2   using namespace kittens::prototype;
3   using namespace kittens::prototype::lcf;
4   template<int M_BLOCK, int N_BLOCK>
5   struct matmul_layout {
6     using  base_tile      = st_bf<64, 64>;
7     using  global_layout  = gl<bf16, 1, 1, -1, -1, base_tile>;
8     struct globals         { global_layout A, B, C; };
9     struct input_block     { base_tile a[M_BLOCK], b[N_BLOCK]; };
10    struct finish_block    { base_tile c[M_BLOCK][N_BLOCK]; };
11    struct common_state    { int2 coord; };
12    struct consumer_state { rt_fl<16, N_BLOCK*base_tile::cols> accum; };
13  };
14  template<int _M_BLOCK=2, int _N_BLOCK=4, int _SUPER_M=12>
15  struct matmul_template {
16    static constexpr int M_BLOCK = _M_BLOCK, N_BLOCK = _N_BLOCK, SUPER_M = _SUPER_M;
17    using layout     = matmul_layout<M_BLOCK, N_BLOCK>;
18    using wide_tile = st_bf<64, 64*N_BLOCK>;
19    static constexpr int NUM_CONSUMER_WARPS=M_BLOCK*4, INPUT_PIPE_STAGES=4,
            ↪ PRODUCER_BARRIER_ARRIVALS=1;
```

```
1    // Helper functions
2    template<bool PERISISTENT_GRID=true> __host__ static inline dim3 grid(int M, int N, int K) {
3      return dim3(PERISISTENT_GRID ? 132 : M*N/(M_BLOCK*N_BLOCK*layout::base_tile::num_elements)
             ↪ );
4    }
5      // ThunderKittens template functions
6    __device__ static inline void common_setup(common_setup_args<layout> args) {
7      int Rblocks = args.globals.C.rows / (M_BLOCK*64), Cblocks = args.globals.C.cols / (N_BLOCK
             ↪ *64);
8      int super_rows = (Rblocks/SUPER_M)*SUPER_M,
9        final_rows = Rblocks - super_rows,
10       super_repeat = SUPER_M*Cblocks;
11     int task_id = args.task_iter*gridDim.x + blockIdx.x;
12     if (task_id < super_rows * Cblocks)
13       args.common.coord = { SUPER_M*(task_id/super_repeat) + task_id%SUPER_M,
14                             (task_id%super_repeat)/SUPER_M };
15     else if (task_id < Rblocks*Cblocks) {
16       int remainder_id = task_id - super_rows*Cblocks;
17       args.common.coord = { super_rows + (remainder_id%final_rows), remainder_id/final_rows };
18     }
19     else { // Id is too high, no more work to do
20       args.num_iters = -1;
21       return;
22     }
23     args.num_iters = args.globals.A.cols/64;
24     int id = warpgroup::groupid() == NUM_CONSUMER_WARPS/4 ? 0 : warpgroup::groupid(); //
             ↪ producer sets as 0
25     args.common.coord = { args.common.coord.x*M_BLOCK + id, args.common.coord.y*N_BLOCK };
26   }
27   struct producer {
28     __device__ static void setup(producer_setup_args<layout> args) {
29       warpgroup::decrease_registers<40>(); // decrease registers for producers
30     }
31     __device__ static void load(producer_load_args<layout> args) {
32       if(warpgroup::warpid() == 0) {
33         tma::expect(args.inputs_arrived, args.input);
34         for(int i = 0; i < M_BLOCK; i++)
35           tma::load_async(args.input.a[i], args.globals.A,
36                           {args.common.coord.x+i, args.iter}, args.inputs_arrived);
37         for(int i = 0; i < N_BLOCK; i++)
38           tma::load_async(args.input.b[i], args.globals.B,
39                           {args.iter, args.common.coord.y+i}, args.inputs_arrived);
40       }
41     }
42   };
43   struct consumer {
44     __device__ static void setup(consumer_setup_args<layout> args) {
45       warpgroup::increase_registers<232>(); // increase registers for consumers
46       zero(args.state.accum);
47     }
48     __device__ static void compute(consumer_compute_args<layout> args) {
49       warpgroup::mma_AB(
50         args.state.accum, // dest registers
51         args.input.a[warpgroup::groupid()], // A matrix
52         reinterpret_cast<wide_tile&>(args.input.b) // B matrix
53       );
54       warpgroup::mma_async_wait();
55       if(laneid() == 0) arrive(args.inputs_finished);
56     }
57     __device__ static void finish(consumer_finish_args<layout> args) {
58       warpgroup::store(reinterpret_cast<wide_tile&>(args.finish.c[warpgroup::groupid()]), args
             ↪ .state.accum);
59       warpgroup::sync();
60       if(warpgroup::warpid() == 0) for(int i = 0; i < N_BLOCK; i++) {
61         tma::store_async(args.globals.C, args.finish.c[warpgroup::groupid()][i],
62                          {args.common.coord.x, args.common.coord.y+i});
63         tma::store_async_read_wait(); // wait that store is finished before reusing finish
                ↪ memory
64       }
65       zero(args.state.accum);
66       if(laneid() == 0) arrive(args.finish_finished);
67     }
68   };
69 };
```

Figure 15: Templated matrix multiply kernel which is reasonably competitive with CuBLAS.

## C.2 LONG CONVOLUTION

This section shows the long convolution kernel for sequence length 4096, written in the TK abstractions. We use the FFT convolution algorithm, computed via Monarch Matrices, for our long convolution kernel (Cooley & Tukey, 1965; Fu et al., 2023a; Dao et al., 2022a).

```
1   struct consumer {
2     __device__ static void setup(consumer_setup_args<layout> args) {
3       warpgroup::consumer_registers<NUM_CONSUMER_WARPS/4>();
4       int iters_per_head = (args.globals.x.batch + NUM_CONSUMER_WARPGROUPS-1) /
              ↪ NUM_CONSUMER_WARPGROUPS;
5       args.state.current_head = (0 / iters_per_head)*132 + blockIdx.x; // start for iter 0
6       using consumers = group<NUM_CONSUMER_WARPS>;
7       consumers::load(args.scratch.f,       args.globals.f,       {0, 0, 0, 0});
8       consumers::load(args.scratch.finv,    args.globals.finv,    {0, 0, 0, 0});
9       consumers::load(args.scratch.tw,      args.globals.tw,      {0, 0, 0, 0});
10      consumers::load(args.scratch.twinv_t, args.globals.twinv_t, {0, 0, 0, 0});
11      load_head_data(args.scratch, args.globals, args.state.current_head);
12    }
13    __device__ static void compute(consumer_compute_args<layout> args) {
14
15      int warpgroupid = warpgroup::warpid()/kittens::WARPGROUP_WARPS;
16      int default_barrer_id = warpgroupid + 4;
17      // X = F^T X
18      crt_fl<16, 64> mma_reg; // 64 registers
19      crt_bf<16, 64> accum, tmp; // 32 registers each
20      warpgroup::mm_AB(mma_reg.real, args.scratch.f.real, args.input.x[warpgroup::groupid()]);
21      warpgroup::mm_AB(mma_reg.imag, args.scratch.f.imag, args.input.x[warpgroup::groupid()]);
22      warpgroup::mma_async_wait();
23      copy(accum, mma_reg);
24      warpgroup::load(tmp, args.scratch.tw); // for twiddle first
25      mul(accum, accum, tmp);
26      group<NUM_CONSUMER_WARPS>::sync(2);
27      warpgroup::mm_AB(mma_reg, accum, args.scratch.f);
28      warpgroup::mma_async_wait();
29      copy(accum, mma_reg);
30      warpgroup::load(tmp, args.scratch.kf); // for filter second
31      mul(accum, accum, tmp);
32      warpgroup::mm_AB(mma_reg, accum, args.scratch.finv);
33      warpgroup::mma_async_wait();
34      copy(accum, mma_reg);
35      warpgroup::load(tmp, args.scratch.twinv_t); // twiddle inverse is pre-transposed
36      mul(accum, accum, tmp);
37      warpgroup::store(args.scratch.tmp[warpgroup::groupid()], accum); // must store for AtB
38      warpgroup::sync(default_barrer_id);
39      warpgroup::mm_AB(mma_reg, args.scratch.finv, args.scratch.tmp[warpgroup::groupid()]);
40      warpgroup::mma_async_wait();
41      warpgroup::store(args.output.o[warpgroup::groupid()], mma_reg.real);
42      warpgroup::sync(default_barrer_id);
43
44      if(laneid() == 0) {
45        arrive(args.inputs_finished);
46        arrive(args.outputs_arrived);
47      }
48      __syncwarp();
49      int iters_per_head = (args.globals.x.batch + NUM_CONSUMER_WARPGROUPS-1) /
              ↪ NUM_CONSUMER_WARPGROUPS;
50      int next_head = ((args.iter+1) / iters_per_head)*132 + blockIdx.x;
51      if(next_head != args.state.current_head) {
52        load_head_data(args.scratch, args.globals, next_head);
53        args.state.current_head = next_head;
54      }
55    }
56    __device__ static void finish(consumer_finish_args<layout> args) { if(laneid() == 0)
            ↪ arrive(args.finish_finished); }
57  };
58 };
```

Figure 16: A convolution kernel for context length 4096, written in the TK LCSF template, which outperforms FlashFFTConv (Fu et al., 2023c).

```cpp
using namespace kittens;
using namespace kittens::prototype;
using namespace kittens::prototype::lcsf;
template<int _wg> struct fftconv_4096_layout { // 4096
  static constexpr int wg = _wg;
  using seq_tile      = st_bf<64, 64>;
  using seq_layout    =    gl<bf16, -1, -1, 64, 64, seq_tile>;
  using filter_layout = cgl<gl<bf16,  1, -1, 64, 64, seq_tile>>;
  using fft_layout    = cgl<gl<bf16,  1,  1, 64, 64>>;
  struct globals {
    seq_layout o, x;
    filter_layout kf;
    fft_layout f, finv, tw, twinv_t;
  };
  struct input_block    { seq_tile x[wg]; };
  struct output_block   { seq_tile o[wg]; };
  struct scratch_block  {
    cst_bf<64, 64> kf, f, finv, tw, twinv_t, tmp[2];
  };
  struct consumer_state { int current_head; };
};
struct fft_4096_template {
  static constexpr int NUM_CONSUMER_WARPS=8, NUM_CONSUMER_WARPGROUPS=NUM_CONSUMER_WARPS/4,
        ↪ NUM_BLOCKS=1, OUTPUT_PIPE_STAGES=2, INPUT_PIPE_STAGES=4;
  using layout = fftconv_4096_layout<NUM_CONSUMER_WARPGROUPS>;
  // mine
  __device__ static inline void load_head_data(typename layout::scratch_block &scratch, const
        ↪ layout::globals &g, int head) {
    using consumers = group<NUM_CONSUMER_WARPS>;
    consumers::sync(3);
    consumers::load(scratch.kf, g.kf, {0, head, 0, 0}); // next chunk
    consumers::sync(3);
  }
  // tk
  __device__ static void common_setup(common_setup_args<layout> args) {
    int heads_handled = (args.globals.x.depth+131-blockIdx.x) / 132; // I am guaranteeing
          ↪ batch is handled by just one block.
    int iters_per_head = (args.globals.x.batch + NUM_CONSUMER_WARPGROUPS-1) /
          ↪ NUM_CONSUMER_WARPGROUPS;
    args.num_iters = args.task_iter == 0 ? heads_handled * iters_per_head : -1;
  }
  struct producer {
    __device__ static void setup(producer_setup_args<layout> args) {
      warpgroup::producer_registers();
    }
    __device__ static void load(producer_load_args<layout> args) {
      int iters_per_head = (args.globals.x.batch + NUM_CONSUMER_WARPGROUPS-1) /
            ↪ NUM_CONSUMER_WARPGROUPS;
      int head  = (args.iter / iters_per_head)*132 + blockIdx.x;
      int batch = (args.iter % iters_per_head) * NUM_CONSUMER_WARPGROUPS;
      if(warpgroup::warpid() == args.iter%4) {
        tma::expect_bytes(args.inputs_arrived, sizeof(args.input.x[0]) * min((int)
              ↪ NUM_CONSUMER_WARPGROUPS, (int)(args.globals.x.batch - batch)));
        for(int b = batch; b < batch+NUM_CONSUMER_WARPGROUPS && b < args.globals.x.batch; b++)
              ↪ {
          tma::load_async(args.input.x[b-batch], args.globals.x, { b, head, 0, 0 }, args.
                ↪ inputs_arrived);
        }
        if(laneid() == 0) arrive(args.inputs_arrived, 3); // extra arrivals needed
        __syncwarp();
      }
    }
    __device__ static void store(producer_store_args<layout> args) {
      int iters_per_head = (args.globals.x.batch + NUM_CONSUMER_WARPGROUPS-1) /
            ↪ NUM_CONSUMER_WARPGROUPS;
      int head  = (args.iter / iters_per_head)*132 + blockIdx.x;
      int batch = (args.iter % iters_per_head) * NUM_CONSUMER_WARPGROUPS;
      if(warpgroup::warpid() == args.iter%4) {
        for(int b = batch; b < batch+NUM_CONSUMER_WARPGROUPS && b < args.globals.x.batch; b++)
              ↪ {
          tma::store_async(args.globals.o, args.output.o[b-batch], { b, head, 0, 0 });
        }
        tma::store_async_read_wait();
        if(laneid() == 0) arrive(args.outputs_finished, 4);
        __syncwarp();
      }
    }
  };
```

## C.3 ATTENTION

This section shows non-causal attention at head dimensions $64, 128$, in the TK abstractions.

```
 1          exp2(args.state.max_vec_last_scaled, args.state.max_vec_last_scaled);
 2          mul(args.state.norm_vec, args.state.norm_vec, args.state.max_vec_last_scaled);
 3          row_sum(args.state.norm_vec, args.state.att_block, args.state.norm_vec); // accumulate
              ↪ onto the norm_vec
 4          mul_row(args.state.o_reg, args.state.o_reg, args.state.max_vec_last_scaled); //
              ↪ normalize o_reg before mma
 5          copy(args.state.att_block_mma, args.state.att_block); // convert to bf16 for mma
 6          // O += A @ V
 7          warpgroup::mma_AB(args.state.o_reg, args.state.att_block_mma, args.input.v);
 8          warpgroup::mma_async_wait();
 9          if(laneid() == 0) arrive(args.inputs_finished); // done!
10        }
11      __device__ static inline void finish(consumer_finish_args<layout> args) {
12          if((args.common.seq*NUM_WORKERS+warpgroup::groupid())*64 >= args.globals.Q.rows) return;
              ↪  // out of bounds?
13          div_row(args.state.o_reg, args.state.o_reg, args.state.norm_vec);
14          auto &o_smem = reinterpret_cast<typename layout::qo_tile&>(args.scratch.q[warpgroup::
              ↪ groupid()]);
15          warpgroup::store(o_smem, args.state.o_reg);
16          warpgroup::sync(warpgroup::groupid());
17          if(warpgroup::warpid() == 0)
18            tma::store_async(args.globals.O, o_smem, {args.common.batch, args.common.head, args.
                ↪ common.seq*NUM_WORKERS+warpgroup::groupid(), 0});
19        }
20    };
21  };
22  // kernel is kittens::prototype::lcf::kernel<attn_fwd_template<HEAD_DIM>>;
```

Figure 17: A templated non-causal attention kernel for head dims. 64 and 128 that competes with FlashAttention-3.

```
1   using namespace kittens;
2   using namespace kittens::prototype;
3   using namespace kittens::prototype::lcf;
4   template<int D, int NUM_WORKERS> struct attn_fwd_layout {
5     using qo_tile   = st_bf<64, D>;
6     using kv_tile   = st_bf<D==64?192:128, D>;
7     using qo_global = kittens::gl<bf16, -1, -1, -1, D, qo_tile>;
8     using kv_global = kittens::gl<bf16, -1, -1, -1, D, kv_tile>;
9     struct globals { qo_global O, Q; kv_global K, V; };
10    struct input_block    { kv_tile k, v; };
11    struct scratch_block  { qo_tile q[NUM_WORKERS]; };
12    struct common_state   { int batch, head, seq; };
13    struct consumer_state {
14      rt_fl<16, qo_tile::cols> o_reg;
15      col_vec<rt_fl<16, kv_tile::rows>> max_vec, norm_vec;
16      col_vec<rt_fl<16, kv_tile::rows>> max_vec_last_scaled, max_vec_scaled;
17      rt_fl<16, kv_tile::rows> att_block;
18      rt_bf<16, kv_tile::rows> att_block_mma;
19    };
20  };
21  template<int D> struct attn_fwd_template {
22    static constexpr int NUM_CONSUMER_WARPS = 12, NUM_WORKERS = NUM_CONSUMER_WARPS/4,
          ↪ INPUT_PIPE_STAGES = 2;
23    using layout = attn_fwd_layout<D, NUM_WORKERS>;
24    __device__ static inline void common_setup(common_setup_args<layout> args) {
25      args.common.batch = blockIdx.z; args.common.head = blockIdx.y; args.common.seq = blockIdx.
            ↪ x;
26      args.num_iters = args.task_iter == 0 ? args.globals.K.rows/layout::kv_tile::rows : -1;
27    }
28    struct producer {
29      __device__ static inline void setup(producer_setup_args<layout> args) {
30        warpgroup::producer_registers();
31      }
32      __device__ static inline void load(producer_load_args<layout> args) {
33        if(warpgroup::warpid() == 0) {
34          tma::expect(args.inputs_arrived, args.input);
35          tma::load_async(args.input.k, args.globals.K, {args.common.batch, args.common.head,
                ↪ args.iter, 0}, args.inputs_arrived);
36          tma::load_async(args.input.v, args.globals.V, {args.common.batch, args.common.head,
                ↪ args.iter, 0}, args.inputs_arrived);
37        }
38        else if(laneid() == 0) arrive(args.inputs_arrived);
39      }
40    };
41    struct consumer {
42      __device__ static inline void setup(consumer_setup_args<layout> args) {
43        warpgroup::consumer_registers<NUM_WORKERS>();
44        if((args.common.seq*NUM_WORKERS + warpgroup::groupid())*layout::qo_tile::rows < args.
              ↪ globals.Q.rows) // out of bounds?
45          warpgroup::load(args.scratch.q[warpgroup::groupid()], args.globals.Q,
46                          {args.common.batch, args.common.head, args.common.seq*NUM_WORKERS+
                              ↪ warpgroup::groupid(), 0});
47        zero(args.state.o_reg);
48        zero(args.state.norm_vec);
49        neg_infty(args.state.max_vec);
50        warpgroup::sync(warpgroup::groupid());
51      }
52      __device__ static inline void compute(consumer_compute_args<layout> args) {
53        constexpr float TEMPERATURE_SCALE = (D == 128) ? 0.08838834764f*1.44269504089f : 0.125f
              ↪ *1.44269504089f;
54        warpgroup::mm_ABt(args.state.att_block,args.scratch.q[warpgroup::groupid()],args.input.k
              ↪ );
55        mul(args.state.max_vec_last_scaled,args.state.max_vec,TEMPERATURE_SCALE);
56        warpgroup::mma_async_wait();
57        // softmax
58
59        row_max(args.state.max_vec, args.state.att_block, args.state.max_vec); // accumulate
              ↪ onto the max_vec
60        mul(args.state.max_vec_scaled, args.state.max_vec, TEMPERATURE_SCALE);
61        mul(args.state.att_block, args.state.att_block, TEMPERATURE_SCALE);
62        sub_row(args.state.att_block, args.state.att_block, args.state.max_vec_scaled);
63        exp2(args.state.att_block, args.state.att_block);
64        sub(args.state.max_vec_last_scaled, args.state.max_vec_last_scaled, args.state.
              ↪ max_vec_scaled);
```

## C.4 Rotary positional encodings

This section shows the rotary kernel for head dimension 128, written in the TK abstractions.

```
1        load(args.state.cos, args.globals.cos, idx);
2      }
3    __device__ static void compute(consumer_compute_args<layout> args) {
4      rt_fl<16, headdim> x;
5      rt_fl<16, headdim/2> x1, x2, temp1, temp2;
6      load(x, args.input.x[warpid()]);
7      if(laneid() == 0) arrive(args.inputs_finished);
8      __syncwarp();
9      for(int i = 0; i < headdim/32; i++) {
10          #pragma unroll
11          for(int j = 0; j < 4; j++) {
12              x1.tiles[0][i].data[j] = x.tiles[0][i].data[j];
13              x2.tiles[0][i].data[j] = x.tiles[0][i+headdim/32].data[j];
14          }
15      }
16      mul(temp1, x1, args.state.cos);
17      mul(temp2, x2, args.state.cos);
18      mul(x2, x2, -1.f);
19      mul(x1, x1, args.state.sin);
20      mul(x2, x2, args.state.sin);
21      add(temp1, temp1, x2);
22      add(temp2, temp2, x1);
23      for(int i = 0; i < headdim/32; i++) {
24          #pragma unroll
25          for(int j = 0; j < 4; j++) {
26              x.tiles[0][i].data[j]            = temp1.tiles[0][i].data[j];
27              x.tiles[0][i+headdim/32].data[j] = temp2.tiles[0][i].data[j];
28          }
29      }
30      store(args.output.o[warpid()], x);
31      __syncwarp();
32      if(laneid() == 0) arrive(args.outputs_arrived);
33    }
34    __device__ static void finish(consumer_finish_args<layout> args) {
35      if(laneid() == 0) arrive(args.finish_finished); // nothing to do here
36    }
37  };
38 };
```

Figure 18: A templated rotary kernel for head dim. 128 that outperforms popular Triton baselines.

```cpp
using namespace kittens;
using namespace kittens::prototype;
using namespace kittens::prototype::lcsf;
template<int _headdim, int _warps> struct rotary_layout {
  static constexpr int headdim = _headdim, warps = _warps;
  using seq_tile    = st_bf<16, headdim>;
  using seq_global  = gl<bf16, -1, -1, -1, headdim, seq_tile>;
  using rope_global = gl<bf16,  1,  1, -1, headdim/2>;
  struct globals {
    seq_global o, x;
    rope_global sin, cos;
    int batches; // how many batches per block, for sizing grid
  };
  struct input_block    { seq_tile x[warps]; };
  struct output_block   { seq_tile o[warps]; };
  struct producer_state { int active_warps;  };
  struct consumer_state { rt_fl<16, headdim/2> sin, cos; }; // long-resident tiles
};
template<int _headdim> struct rotary_template {
  static constexpr int headdim=_headdim, NUM_CONSUMER_WARPS=8, NUM_BLOCKS=1,
        ↪ OUTPUT_PIPE_STAGES=3, INPUT_PIPE_STAGES=3;
  using layout = rotary_layout<headdim, NUM_CONSUMER_WARPS>;
  __device__ static inline void common_setup(common_setup_args<layout> args) {
    if(args.task_iter == 0) {
      args.num_iters = min(args.globals.batches, (int)(args.globals.x.batch-blockIdx.y*args.
            ↪ globals.batches)) * args.globals.x.depth; // batches*heads handled by block
    }
    else args.num_iters = -1;
  }
  struct producer {
    __device__ static void setup(producer_setup_args<layout> args) {
      warpgroup::producer_registers();
      args.state.active_warps = min((int)NUM_CONSUMER_WARPS,
                                    (int)(args.globals.x.rows/16 - blockIdx.x*
                                          ↪ NUM_CONSUMER_WARPS));
    }
    __device__ static void load(producer_load_args<layout> args) {
      if(warpgroup::warpid() == args.iter%4) {
          kittens::coord idx = { blockIdx.y*args.globals.batches+args.iter/args.globals.x.
                ↪ depth,
                                args.iter%args.globals.x.depth,
                                blockIdx.x*NUM_CONSUMER_WARPS,
                                0 };
          tma::expect_bytes(args.inputs_arrived, sizeof(layout::seq_tile)*args.state.
                ↪ active_warps);
          for(int i = 0; i < args.state.active_warps; i++) {
              tma::load_async(args.input.x[i], args.globals.x, {idx.b,idx.d,idx.r+i,idx.c},
                    ↪ args.inputs_arrived);
          }
          if(laneid() == 0) arrive(args.inputs_arrived, 3);
          __syncwarp();
      }
    }
    __device__ static void store(producer_store_args<layout> args) {
      if(warpgroup::warpid() == args.iter%4) {
          kittens::coord idx = { blockIdx.y*args.globals.batches+args.iter/args.globals.x.
                ↪ depth,
                                args.iter%args.globals.x.depth,
                                blockIdx.x*NUM_CONSUMER_WARPS,
                                0 };
          for(int i = 0; i < args.state.active_warps; i++) {
              tma::store_async(args.globals.o, args.output.o[i], {idx.b,idx.d,idx.r+i,idx.c});
          }
          tma::store_async_read_wait();
          if(laneid() == 0) arrive(args.outputs_finished, 4);
          __syncwarp();
      }
    }
  };
  struct consumer {
    __device__ static void setup(consumer_setup_args<layout> args) {
      warpgroup::consumer_registers<NUM_CONSUMER_WARPS/4>();
      kittens::coord idx = { blockIdx.x*NUM_CONSUMER_WARPS + warpid(), 0 };
      load(args.state.sin, args.globals.sin, idx); // could be better coalesced but doing just
            ↪ once
```

# D  LIBRARY IMPLEMENTATION DETAILS

This section provides additional implementation details for THUNDERKITTENS.

## D.1  TILE DATA STRUCTURES

The core primitive in THUNDERKITTENS is the tile data structure as introduced in section 3.1. Tiles exist at the shared memory and register memory levels of the GPU hierarchy, and are created with multiples of $16 \times 16$ in dimension.

**Precision**  THUNDERKITTENS is extensible across data types: FP32, FP16, BF16, and FP8. Designing a unified tile data structure that seamlessly supports different data types is challenging for two reasons:

1. Each data type requires using different memory layouts both at the shared and register memory levels, in order to use specialized hardware instructions like tensor cores. [3]
2. Each data type uses a different amount of space, meaning that the $16 \times 16$ tile that the user sees could contain a fixed number of bits, fixed number of elements, or some other option. Ideally, we can store elements in fully packed formats (e.g., `bf16_2` for `bf16`, `e4m3_8x4` for `e4m3` FP8).

We let the users think in terms of the number of elements per tile. When the user defines a $16 \times 32$ tile for instance, we store this as $16 \times 8$ packed elements of `e4m3_8x4` or $16 \times 16$ packed elements of `bf16_2` in registers. In the library, we define and operate an *underlying tile width* for the tiles to hide this complexity from the user. Taking care of differences across data types at tile data structure level, we can then use the exact same library functions (e.g., `mma`, `exp`, `cumsum`) across tiles of different data types, preserving the simplicity of the library.

**Padding**  Some AI workloads require shapes that are *not* multiples of 16. THUNDERKITTENS provides mechanisms to support these workloads, too, without compromising performance on hardware-friendly workloads.

1. Loads & Stores. THUNDERKITTENS loads and stores take an optional template arguments for whether to assume tensors are multiples of 16. If not, each load or store is preceded by a check to ensure that it is in-bounds. Out-of-bound loads are filled with zeros; out-of-bound stores are not performed. For safety, these checks are enabled by default; however, they are not free, and they can also be disabled by setting the appropriate template flag. This abides by TK's philosophy of "extensible, but with good defaults." For TMA loads and stores, we use the built-in hardware padding features (that is, out-of-bounds accesses are automatically filled in with zeros).
2. Fills & Masks. In addition to preventing illegal memory accesses, one often needs to alter data within a tile to prevent accidental computations. THUNDERKITTENS provides functionality for this, too, in the form of six functions: top_fill, bottom_fill, left_fill, right_fill, triu, and tril, which respectively fill the top, bottom, left, right, upper triangle, or lower triangle of tiles. In our experience, these functions have proven sufficient for all kernels considered.

## D.2  SHARED MEMORY LAYOUTS

To illustrate some of the choices available in shared memory layouts, this appendix outlines six different shared memory layouts for GPU tiles: a naive row-major layout, a padded layout, a simple swizzled layout, and three more specialized swizzled layouts. We are particularly interested in which memory *banks* (numbered from 00 to 31) store each element of the tile; for each layout, we color and label the element of the tile accordingly. We illustrate all layouts using a $32 \times 64$ 16-bit tile.

## D.3  NAIVE LAYOUT

A row-major layout, illustrated in figure 19, is among the simplest layouts. It has the benefit of accurately reflecting tensor layouts in HBM. Furthermore, for access patterns that access row-wise, it has no bank conflicts. But when loading or storing tensor core register layouts, it suffers 8-way bank conflicts, and is thus extremely slow.

---

[3] `https://docs.nvidia.com/cuda/parallel-thread-execution/` `#asynchronous-warpgroup-level-matrix-instructions`

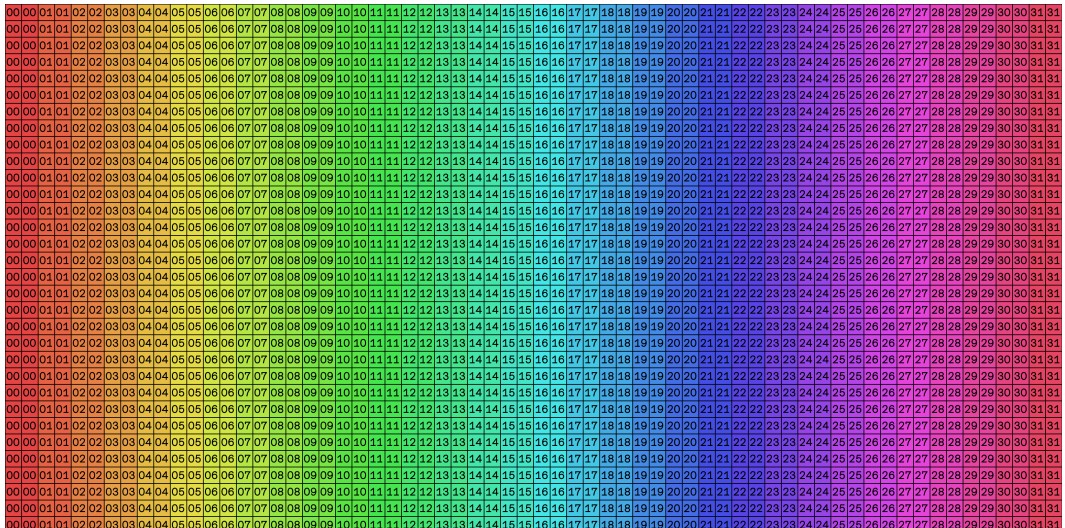

Figure 19: Row-major shared memory layout.

```
1  bf16* naive_layout(bf16 *data, int r, int c) {
2      return &data[r * columns + c];
3  }
```

### D.3.1 PADDED LAYOUT

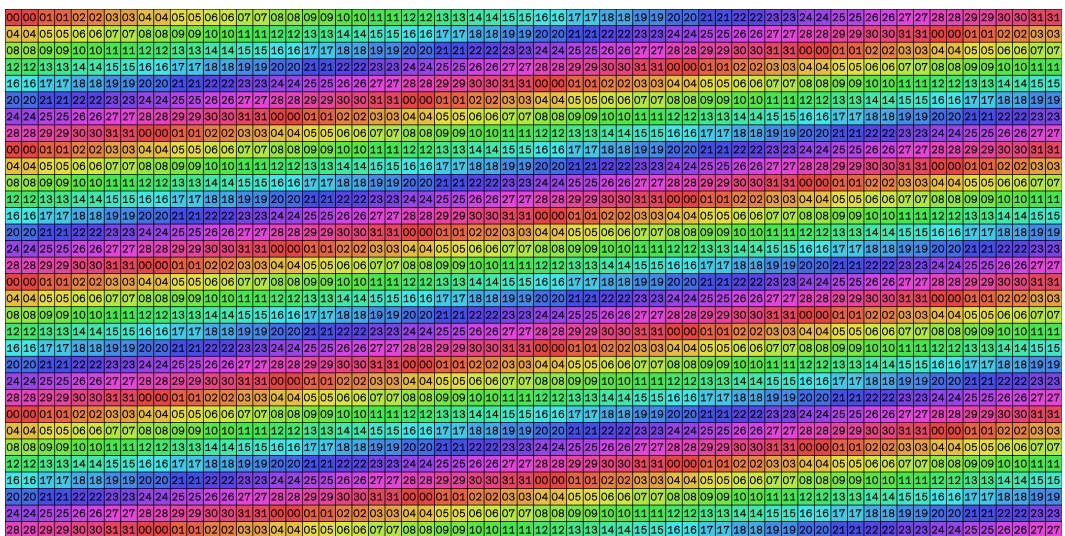

Figure 20: Padded shared memory layout.

A common solution to these bank conflicts is to "pad" each row by one memory bank, thereby introducing an offset to shift consecutive elements of a column into different memory banks. This eliminates bank conflicts, but creates misaligned addresses which interferes with fast instructions that require aligned addresses. For example, it wouldn't be possible to use TMA to store the second row of this layout due to it only having a 16-byte alignment, whereas TMA requires 128-byte alignments.

```
1  bf16* padded_layout(bf16 *data, int r, int c) {
2      return &data[r * (columns+1) + c];
3  }
```

### D.3.2 NAIVE SWIZZLED LAYOUT

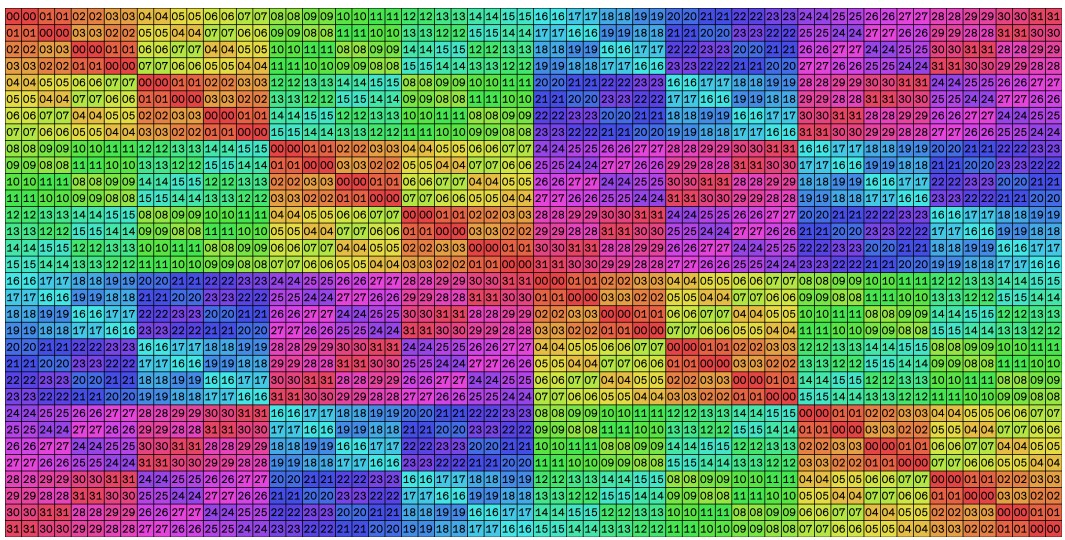

Figure 21: Naive swizzled shared memory layout.

A third option is to "swizzle" the memory, in which progressive rows are reshuffled to alter their banking. This layout accomplishes this by xor'ing the index with the row, which reduces bank conflicts. However, this layout lacks hardware support for HGMMA and UTMA instructions, which are particularly important on H100 GPUs for achieving high performance. Additionally, the granularity of the swizzling must be large enough to totally prevent bank conflicts when loading into registers. We illustrate a simple, naive swizzling pattern here, which used to be recommended for preventing bank conflicts before the advent of tensor cores:

```
1  bf16* row_swizzled_layout(bf16 *data, int r, int c) {
2      uint64_t addr = (uint64_t)&data[r * columns + c];
3      return (bf16*)(addr ^ (r << 2));
4  }
```

### D.3.3    32 BYTE SWIZZLING

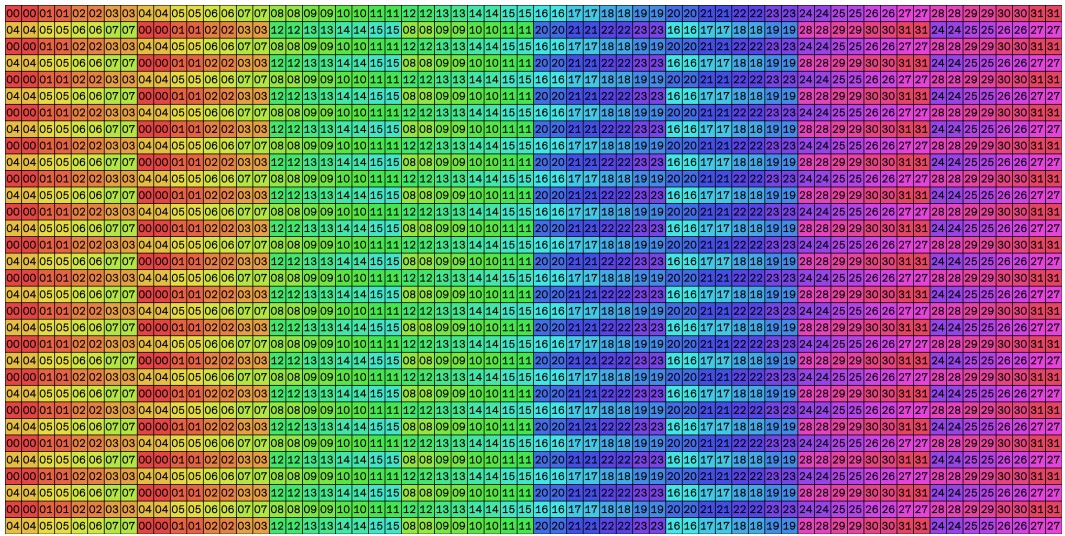

Figure 22: 32 byte swizzled shared memory layout.

32 byte swizzling is the first of a family of layouts (of which we will examine three), where instead of swizzling the index with the row, the memory address is instead swizzled directly with itself. This layout is defined by the following C code:

```
bf16* swizzled_layout_32B(bf16 *data, int r, int c) {
    uint64_t addr = (uint64_t)&data[r * columns + c];
    return (bf16*)(addr ^ (((addr % (32*8)) >> 7) << 4));
}
```

This layout here suffers from 4-way bank conflicts, but is valid for all tiles whose width is a multiple of 16. However, importantly, it has (as do its siblings below) hardware support from both `HGMMA` and `UTMA` instructions.

### D.3.4  64 BYTE SWIZZLING

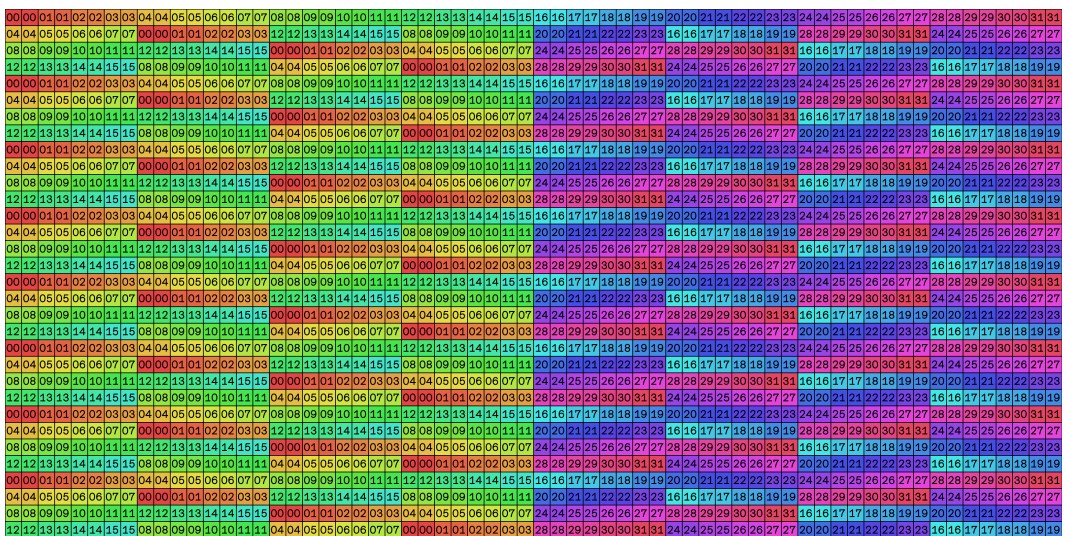

Figure 23: 64 byte swizzled shared memory layout.

64 byte swizzling is a layout similar to 32 byte swizzling with a more aggressive pattern:

```
bf16* swizzled_layout_64B(bf16 *data, int r, int c) {
    uint64_t addr = (uint64_t)&data[r * columns + c];
    return (bf16*)(addr ^ (((addr % (64*8)) >> 7) << 4));
}
```

64 byte swizzling suffers from just 2-way bank conflicts, but is only valid for tiles whose width is a multiple of 32 (for half-precision types, or 16 for full-precision).

### D.3.5  128 BYTE SWIZZLING.

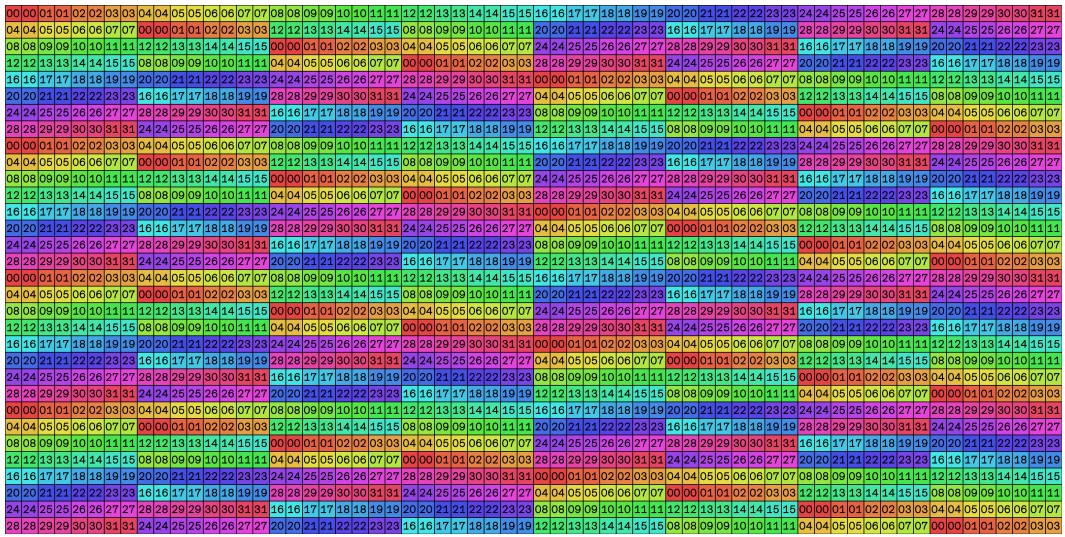

Figure 24: 128 byte swizzled shared memory layout.

128 byte swizzling is a further extension of its kin:

```
1  bf16* swizzled_layout_128B(bf16 *data, int r, int c) {
2      uint64_t addr = (uint64_t)&data[r * columns + c];
3      return (bf16*)(addr ^ (((addr % (128*8)) >> 7) << 4));
4  }
```

Finally, 128 byte swizzling has no bank conflicts, but is only valid for half-precision tiles whose width is a multiple of 64.

### D.3.6    THUNDERKITTENS APPROACH

After substantial evaluation of these layouts, we concluded that the three final layouts were the three most important, because HGMMA and UTMA instructions are critical to high performance, and furthermore that they are good enough to yield high performance across many kernels. Correspondingly, depending on the width of the tile at compile time we select the highest level of swizzling possible to minimize bank conflicts.

