# OpenReview forum: "ThunderKittens: Simple, Fast, and $\textit{Adorable}$ Kernels"
_ICLR.cc/2025/Conference — ICLR 2025 Spotlight_

### Official Review · Reviewer_5GXe · 2024-10-31

**Soundness:** 3
**Presentation:** 3
**Contribution:** 3
**Rating:** 6
**Confidence:** 4

**Summary:**

The paper introduces THUNDERKITTENS (TK), a framework that simplifies writing AI kernels for GPUs while still allowing for high performance. Using a few key abstractions, TK provides tools for developers to create efficient kernels without deep expertise in GPU programming. Through benchmarking, the authors show that TK performs on par with or better than other leading frameworks like CuBLAS and FlashAttention-3 for various AI tasks. TK’s accessible design, inspired by PyTorch and NumPy, aims to make high-performance kernel development more straightforward and accessible to a wider audience.

**Strengths:**

The paper offers a fresh and practical approach to GPU kernel programming, using only a handful of essential abstractions to make high-performance kernel writing accessible to a wider range of developers. This simplicity-oriented approach can reduce the complexity typically associated with GPU development, which could be particularly valuable for those without extensive CUDA experience. In terms of performance, THUNDERKITTENS shows impressive results, even surpassing established libraries like CuBLAS and FlashAttention-3 in several tasks, especially in backward pass operations for attention mechanisms and linear attention. The results strongly suggest that TK’s design strikes a good balance between simplicity and performance optimization. Furthermore, by aligning its design with PyTorch and NumPy, TK makes it easier for non-specialists to adopt, potentially expanding the accessibility of efficient GPU programming.

**Weaknesses:**

1- While the minimalistic design is a key strength, it may also limit TK’s flexibility for more specialized AI tasks that require tailored optimization strategies. As demands grow for handling complex and emerging AI workloads, the current set of abstractions could potentially fall short.

2- The focus on NVIDIA’s H100 GPUs raises questions about how well TK can transfer to other platforms, such as AMD or Apple GPUs. Expanding on cross-platform compatibility would provide more clarity about TK’s broader usability.

3- Though the paper demonstrates strong performance on medium-sized data, it is less clear how TK handles scalability with very large datasets or highly parallelized scenarios. Addressing its limitations in these settings could further support TK’s value in real-world applications.

**Questions:**

Could the authors elaborate on the potential for cross-platform compatibility? Given the focus on NVIDIA’s H100 GPUs, it would be helpful to understand whether TK’s abstractions could be adapted to other GPU architectures, like AMD or Apple, and what challenges might arise.

The paper demonstrates TK’s strong performance on medium-sized data blocks, but could the authors provide more insights into how well TK scales with very large datasets? Are there specific limitations to consider for applications requiring high parallelization or extensive data handling?

Could the authors expand on their design choice to limit TK to a few key abstractions? Are there specific reasons why additional templates or adaptive features were not incorporated, and would doing so have risked undermining the framework’s simplicity?

In scenarios with high memory demands, how does TK manage the balance between memory overhead and computational efficiency? Further detail on this balance could clarify TK’s suitability for applications with varied memory and compute requirements.

Lastly, could the authors clarify TK’s debugging process, especially for users who may not be familiar with GPU optimization? Since GPU kernel errors can be challenging to diagnose, any insights into how TK might support error handling and debugging would be valuable for potential adopters.

---

> ### Author Response · Authors · 2024-11-22
> **Response to review**
>
> Thank you for your positive review of our work! We are glad that you find our approach "fresh" and "valuable" and results "impressive". We carefully address your outstanding questions in our response.
>
> ## Addressing weakness 1. TK’s flexibility for complex AI workloads.
> We focus on a research question: What are the tradeoffs between programming complexity and the accessibility of peak hardware performance? We view the fact that TK is uses minimal primitives, yet provides competitive or higher performance as prior kernels, as a strength rather than a limitation. We provide the summary table below (and in Appendix B.3), showing that:
> 1. TK competes with or outperforms the baselines across settings.
> 2. TK uses a similar amount of lines of code per kernel as Triton, and fewer lines than the other CUDA reference kernels
>
> **Table: Lines of code across TK H100 kernels, state of the art non TK kernels, and the TK speed up over the reference kernels.**
>
> | Workload | TK kernel (lines of code) | Reference kernel (lines of code) | Speed up (max-min) |
> |----------|--------------------------|--------------------------------|-------------------|
> | Attention forwards | 217 | 2325 (CUTLASS FA3) | 0.87-1.14x |
> | GEMM | 84 | 463 (CUTLASS) | 0.98-2.05x |
> | Convolution (N=4096) | 131 | 642 (CUDA FlashFFTConv) | 4.7x |
> | Based linear attention | 282 | 89 (Triton) | 3.7-14.5x |
> | Hedgehog linear attention | 316 | 104 (Triton) | 4.0-6.5x |
> | Mamba-2 | 192 | 532 (Triton) | 3.0-3.7x |
> | Rotary | 101 | 119 (Triton) | 1.1-2.3x |
> | Fused layer norm | 146 | 124 (Triton) | 1.0-2.2x |
>
> Overall, our core abstractions handle the optimization patterns needed for most AI kernels (as demonstrated in our case studies), TK is explicitly designed to be extensible when specialized optimizations are needed. Users can seamlessly mix TK's high-level primitives with custom CUDA code - as we show in our Mamba-2 implementation. This hybrid approach means users get the benefits of TK's abstractions for the bulk of their kernel (typically 95% of the effort), while retaining the flexibility to implement specialized operations where needed.
>
> Philosophically, TK focuses on providing well-chosen defaults that compose well with both each other and custom code. This design ensures TK remains practical and maintainable (ThunderKittens’ source is just a few thousand lines of code) while supporting the full spectrum of AI workloads.
>
> ## Addressing weakness 2. TK’s portability across hardware platforms.
> To address this concern, we provide 2 new kernels for NVIDIA 4090 GPUs and 3 new kernels for Apple M2 chips, to demonstrate that the TK framework ports across hardware platforms. We have added results and discussion for these kernels in Appendix B.
>
> **Table: 4090 Attention FWD Performance (non-causal, batch=16, heads=16)**
>
> | Sequence Length | TK Attention FWD (head dim 64) | TK Attention FWD (non-causal, head dim 128) |
> |----------------|-------------------------------------------------------|--------------------------------------------------------|
> | 1024           | 150 TFLOPs                                            | 141 TFLOPs                                              |
> | 2048           | 154 TFLOPs                                            | 145 TFLOPs                                              |
> | 4096           | 157 TFLOPs                                            | 156 TFLOPs                                              |
> | 8192           | 160 TFLOPs                                            | 148 TFLOPs                                              |
>
> **Table: Apple M2 Standard Attention FWD vs Apple MLX (non-causal, batch=16, heads=16)**
>
> | Sequence Length | TK Attention FWD (head dim 64) | TK Attention FWD (head dim 128) |
> |----------------|--------------------------------|--------------------------------|
> | 512            | 3523.46 vs 3088.41 GFLOPS      | 3263.38 vs 3770.52 GFLOPS      |
> | 1024           | 3723.83 vs 3276.87 GFLOPS      | 3435.89 vs 3977.23 GFLOPS      |
> | 1536           | 3761.81 vs 3313.16 GFLOPS      | 3490.66 vs 4053.37 GFLOPS      |
> | 2048           | 3784.12 vs 3309.63 GFLOPS      | 3488.09 vs 4005.99 GFLOPS      |
> | 2560           | 3793.42 vs 3329.78 GFLOPS      | 3483.83 vs 4047.90 GFLOPS      |
>
> **Table: Apple M2 GEMM TK vs Apple MLX Performance**
>
> | Sequence Length | Performance (GFLOPS)           |
> |----------------|--------------------------------|
> | 1024           | 3830.83 vs 3444.65            |
> | 2048           | 5238.84 vs 4839.45            |
> | 4096           | 5600.58 vs 5190.06            |
> | 8192           | 5675.82 vs 5182.69            |
> | 16384          | 5266.97 vs 5117.65            |
>
> We originally highlighted NVIDIA H100s since they represent the trend of AI hardware growing more complex, with an increasing number of specialized instructions, levels of parallel execution, and opportunities for asynchronous execution.

---

> > ### Author Response · Authors · 2024-11-22
> > **Response to review [continued]**
> >
> > ## Addressing weakness 3. TK’s performance at scale and in parallelized scenarios.
> > We appreciate this suggestion. While multi-GPU parallelization is indeed important for large-scale training, our current focus is on optimizing single-GPU kernel performance. Users of our framework can currently handle multi-GPU communication through established libraries like NCCL, which provides highly optimized primitives for inter-GPU collective operations.
> >
> > We agree that extending the producer-consumer model to multi-GPU contexts is an interesting direction. We are excited to explore such directions in future work–navigating both the interaction between our kernel scheduling and inter-GPU communication patterns, as well as how to balance computation and communication overhead across different GPU topologies.
> >
> > ## Addressing additional questions
> >
> > **Balancing memory overheads and computational efficiency:**
> >
> > We provide a number of opportunities to help the user balance memory and compute efficiency:
> > 1. *Tile sizes and managed memory.* Kernels operate on data in small tiles, due to the limited amount of fast shared and register memory. Developers generally need to tune the tile size to balance the memory use and compute per tile. TK templates all library functions and memory layouts around the tile size, making it easier for users to tune the sizes, without requiring significant kernel rewriting.
> > 2. *Tuning pipelines and occupancy.* More pipeline stages and higher occupancy increases the kernel’s memory demand, but can increase the hardware utilization. We developers users to tune the number of pipeline stages and the occupancy level by modifying a single number.
> >
> > **Debugging process for TK:**
> >
> > The TK development and debugging process includes the following:
> >
> > 1.*Debugging correctness.* First, users write a test case in Python for the kernel that they want, which is used to test kernel correctness. To help users, we provide several examples in our repository for how to hook these Python test cases into the kernel development process.
> > 2. *Debugging compilation errors.* For each library function, we consistently provide static checks for the inputs and outputs that the user passes in. These informative checks tell the user where their mistakes occur. We discuss these checks in lines 245-248 of our original submission.
> > 3. *Debugging performance.* Once we have a correct, compiled kernel, we typically use NVIDIA NSIGHT COMPUTE, an open-source profiling tool that helps users understand where performance bottlenecks occur.
> >
> >
> > We hope our review has addressed your remaining questions. Please let us know if we can provide anything else that would be helpful!

---

> > > ### Author Response · Authors · 2024-11-25
> > > **Following up**
> > >
> > > Dear Reviewer 5GXe,
> > >
> > > Thank you again for your review! We are wondering if our response has addressed your concerns? Please let us know if we can provide anything more and we look forward to your response!

---

### Official Review · Reviewer_ANEJ · 2024-10-31

**Soundness:** 3
**Presentation:** 4
**Contribution:** 3
**Rating:** 8
**Confidence:** 5

**Summary:**

This paper presents ThunderKittens (TK), a C++ embedded library for writing high-performance CUDA kernels for NVIDIA GPUs. It introduces warp-, thread-block-, and grid-level abstractions to facilitate mapping of kernels to the GPU hierarchy. Experimental results indicate that TK can outperform strong industrial baselines, achieving superior performance for GEMM and attention kernels.

**Strengths:**

1. The TK library provides a useful abstraction for writing high-performance asynchronous kernels on GPU.
2. The presentation is clear and accessible, especially the introductory sections on GPU architecture, which provide a helpful overview for ML researchers who may lack in-depth experience with GPU programming.
3. The experimental results are compelling, showing performance on par or better than highly optimized kernels, such as FlashAttention3. The paper also demonstrates significant speedups across different kernel types compared to state-of-the-art frameworks like Triton and PyTorch.

**Weaknesses:**

1. The TK library is still too low-level with too many details, which requires users to manage synchronization carefully and does not simplify the programming burden.
2. The novelty and advantages of TK over CUTLASS are unclear. Many functionalities seem achievable with CUTLASS as well. The authors mention that TK addresses bank conflicts, but the evidence presented is minimal. There appear to be no inherent limitations in CUTLASS that would prevent it from avoiding bank conflicts.
3. Similarly, the benefits of TK over Triton are not well established. Triton, embedded in Python with a PyTorch-like API, may offer a more accessible interface. By contrast, TK, embedded in C++, still requires explicit handling of communication with mbarrier operations like expect and arrive. No user study or lines of code comparisons are provided to demonstrate that TK improves programmer productivity.
4. Experimental results are good, but still missing comparisons in some important cases like quantized kernels and causal attention.
5. The work reads more like a system paper, with limited ML-focused insights, raising questions about its fit for ICLR.

Minor:
- P4: "Since the frameworks are not C++ embedded, it can be challenging to use specialized hardware instructions" This statement is inaccurate; TVM provides mechanisms to incorporate low-level TensorCore instructions, and Triton also has [inline](https://triton-lang.org/main/python-api/triton.language.html#inline-assembly) operation to include PTX code.
- Section 2 does not discuss the Tensor Memory Accelerator (TMA) on Hopper, which is essential for asynchronous optimizations mentioned in Contribution 2.
- Appendix B labels appear broken (??).

**Questions:**

1. What are the fundamental challenges preventing CUTLASS from avoiding bank conflicts? Could it be that the FlashAttention3 kernel simply did not select the optimal layout?
2. CUTLASS has implemented both ping-pong and cooperative kernel variants for GEMM, with varying performance across different scenarios. How does TK support ping-pong and cooperative kernels, and could you include a comparison with CUTLASS in Figure 7’s GEMM kernel results?
3. TK appears designed specifically for the Hopper architecture with asynchronous features. Is it also compatible with Ampere or other GPU generations? How does TK’s performance on an A100 compare to Triton?
4. Following Q3, if Blackwell GPUs were released, would TK’s abstractions remain applicable? How do you plan to ensure extensibility across GPU generations?
5. What's the usage of the cost model in Section 2.2? This formula is highly simplified and does not guide any optimization or automatic search later.
6. Section 3.1 discusses various layouts — do users need to manually manage data organization and specify layouts in TK?
7. Figure 5 is just some wrappers of mbarriers. Any insights here?
8. Can TK effectively handle quantized kernels, where data layout is crucial for efficient transfers from TMA and WGMMA computation? How does it perform on FP8 GEMM and FlashAttention kernels?
9. What is TK's performance on causal attention kernels?
10. Please provide detailed experimental configurations in the Appendix. For example, which versions of PyTorch and Triton were used? Was `torch.compile` employed to optimize those network layers? For cuBLAS, was the latest [cuBLASLt](https://developer.nvidia.com/blog/introducing-grouped-gemm-apis-in-cublas-and-more-performance-updates/) autotuning enabled? Since PyTorch also uses Triton as a backend, what distinguishes the two baselines in Figure 8?

---

> ### Author Response · Authors · 2024-11-22
> **Response to review**
>
> Thank you for your thorough review of our work! We are grateful for your detailed feedback, which significantly helped us improve our work. We provide our response to your feedback below.
>
> ## Addressing weakness 1: Simplicity of programming with TK, and Addressing weakness 3: Lines of code comparisons.
> The review mentions that TK is too low-level, which may not simplify the programming burden. Our results and analysis suggest that TK is quite helpful. We provide the summary table below (and in Appendix B.3), showing that:
> 1. TK competes with or outperforms the baselines across settings.
> 2. TK uses a similar amount of lines of code per kernel as Triton, and fewer lines than the other CUDA reference kernels
>
> **Table: Lines of code across TK H100 kernels, state of the art non TK kernels, and the TK speed up over the reference kernels**
> | Workload | TK kernel (lines of code) | Reference kernel (lines of code) | Speed up (max-min) |
> |----------|--------------------------|--------------------------------|-------------------|
> | Attention forwards | 217 | 2325 (CUTLASS FA3) | 0.87-1.14x |
> | GEMM | 84 | 463 (CUTLASS) | 0.98-2.05x |
> | Convolution (N=4096) | 131 | 642 (CUDA FlashFFTConv) | 4.7x |
> | Based linear attention | 282 | 89 (Triton) | 3.7-14.5x |
> | Hedgehog linear attention | 316 | 104 (Triton) | 4.0-6.5x |
> | Mamba-2 | 192 | 532 (Triton) | 3.0-3.7x |
> | Rotary | 101 | 119 (Triton) | 1.1-2.3x |
> | Fused layer norm | 146 | 124 (Triton) | 1.0-2.2x |
>
>
> Additionally, we reiterate that TK manages the following for the user:
> 1. **Warp-level:** We automatically manage memory layouts for the user. We provide optimized library functions inspired by PyTorch (mma, exp, cumsum) to uplevel the programming experience. We provide tile data structures for both shared and register memory (and we manage data type conversions, broadcasts, templating functions around the tile sizes, loads and stores across the memory hierarchy) to help users manage their memory resource utilization.
> 2. **Block-level:** We provide a general kernel template that helps users utilize two important asynchronous execution patterns that are helpful in AI workloads: (1) multi-stage pipeline and (2) producer consumer worker specialization. We set these patterns up for the user.
> We think that giving users the option to set barriers is important for enabling peak performance. Triton does not offer this, and we show in our paper that asynchrony is important. Our goal is to simplify the primitives, without sacrificing performance.
> 3. **Grid-level:** We provide persistent grid support and a function to help users set the thread block launch pattern, which influences L2 reuse.
>
>
> ## Addressing weaknesses 2: Novelty of TK compared to prior frameworks
> We focus on a research question: What are the tradeoffs between programming complexity and the accessibility of peak hardware performance? Our novelty is in showing a new point on the tradeoff space exists. We can use fewer primitives to achieve higher performance on AI workloads, compared to what’s been demonstrated in prior frameworks. Identifying the primitives is non-trivial given the complexity of GPUs, and this is core to our contribution.
>
> To address your feedback, we have included new analysis in Appendix B.3, we believe it has helped the paper and thank you for your suggestion. Please also see this new content below and in our common response. We compare the features supported by different frameworks:
>
> | Feature | TK | CUTLASS | Triton |
> |---------|----|---------| -------|
> | Direct register memory control | YES | YES | NO |
> | Fine-grained asynchronous execution | YES | YES | NO |
> | PyTorch-like library functions | YES | NO | YES |
> | Supports multiple hardware platforms | YES | NO | YES |
>
> We recognize that the following is an imperfect metric. However, we include the sizes of various CUDA libraries below as a rough proxy.  For CUTLASS and TK we report the size of the “include/” directory, and for Triton we report the combined size of the “include/” directories in Triton plus the “include/” in the core MLIR compiler dependency.
>
> | Library  | Size (Bytes) | Date/Version |
> |----------|--------------|--------------|
> | CUTLASS  | 22 MB       | 10/22/2024   |
> | Triton   | 12.6 MB     | 10/22/2024   |
> | TK       | < 1.0 MB    | 10/22/2024   |
>
>
> ## Addressing weakness 4. Additional kernels highlighting the scope of TK's features
> The review notes that our experiments miss some useful kernels. We provide new results for the requested kernels: quantized FP8 GEMM, causal attention.  Please find these results in Appendix B2 and in the common response. We hope that these results address any remaining concerns on experimental validation.

---

> ### Author Response · Authors · 2024-11-22
> **Response to review [continued]**
>
> ## Addressing weakness 5. Highlighting TK’s value to the ML audience
>
> While TK advances systems capabilities, its impact lies in enabling the next wave of ML innovation. As recognized in the ICLR 2025 call for papers, implementation efficiency and hardware optimization are no longer secondary concerns - they are fundamental bottlenecks in advancing AI research.
>
> Our work offers three key contributions that directly address these bottlenecks:
>
> 1. **Democratizing Hardware-Efficient ML Research.** The complexity of hardware optimization has become a critical barrier to ML architecture innovation. TK dramatically lowers this barrier, enabling researchers to rapidly prototype and evaluate novel neural architectures that would previously have required months of specialized CUDA engineering. For instance, our framework makes tensor core optimization accessible through simple interfaces (stated on lines 88-91, 270-280 of our submission).
>
> 2. **Making New ML Architectures Practical.** History shows that major ML advances come from scaling - but scaling requires efficiency. Our framework has already revealed that architectures previously dismissed as impractical can be transformed into state-of-the-art approaches through proper optimization. For example, our implementation shows that linear attention can outperform standard attention in wall-clock time at much shorter sequences than previously thought possible, fundamentally changing the calculus of architecture design decisions. This is just one example of how TK can unlock new families of architectures that were previously considered computationally infeasible.
>
> 3. **Quantitative Impact on ML Workloads.** We demonstrate substantial improvements across core ML operations:
> - $10-40\%$ faster attention computation
> - $8\times$ acceleration for long-range convolutions
> - $6\times-14\times$ speedup for linear attention variants These improvements directly enable research into larger contexts, alternative attention mechanisms, and novel architecture designs.
>
> In our view, these optimizations represent the difference between architectures that can scale and those that cannot. Just as FlashAttention's impact went far beyond its speedup numbers to enable the current wave of large language models, TK aims to unlock the next generation of ML architectures by removing critical implementation barriers that currently constrain innovation.
>
> ## Addressing question 1. Bank conflicts in FlashAttention-3.
> CUTLASS and TK support the exact same capabilities, which is stated on lines 200-202 of our submission. We did not state that CUTLASS faces fundamental challenges causing bank conflicts. We simply observe that bank conflicts are avoidable and the user does not need to be burdened in thinking about layouts, as stated on lines 206-208 of our submission. TK handles the layout selection for the user to help minimize such issues.
>
> ##  Addressing question 2. CUTLASS ping pong and cooperative kernels.
> ThunderKittens supports ping-pong kernels like CUTLASS -- our work has not emphasized ping-pong style kernels, as we have found that it adds considerable complexity for marginal performance improvements. Our GEMM kernel is also cooperative in nature; we also have threads exchange data through shared memory before performing coalesced stores to global memory. We have added comparison to CUTLASS GEMM in Figure 7, and also provide an example ThunderKittens ping-pong GEMM kernel in the appendix which makes further use of the asynchronous tensor core instructions. We find it achieves an additional 0.4% performance above the kernel proposed in the main paper.
>
> ##  Addressing question 3. Extensibility of TK’s ideas across hardware platforms
> The review mentions that our original set of kernels are for H100 GPUs. We provide 2 new kernels for NVIDIA 4090 GPUs and 3 new kernels for Apple M2 chips, to demonstrate that the TK framework ports across hardware platforms. We have added results and discussion for these kernels in Appendix B3.
>
> ## Addressing question 4. Extensibility to future GPU generations
> Thank you for this thoughtful question! While it is difficult to exactly know what changes will be required, since we do not have Blackwell GPUs, we are optimistic that TK’s approach will extend. Our primitives are designed around fundamental hardware properties – e.g., for memory, shared memory is banked, we need coalesced HBM loads, register utilization needs to be carefully managed; e.g., for compute, hardware has multiple execution units that benefit from occupancy, latencies need to be hidden via asynchronous execution. These are not platform specific properties; these fundamentals have already cleanly transferred across hardware platforms and vendors (NVIDIA 4090; NVIDIA H100; Apple M2). We expect the trend to continue.

---

> > ### Author Response · Authors · 2024-11-22
> > **Response to review [continued]**
> >
> > ## Addressing question 5: Section 2.2 helps summarize the kernel costs
> > Thank you for your feedback about Section 2.2. The goal of Section 2.2 is to summarize the various costs in a concise way. As you mention, Section 2 ``helps ML researchers who may lack in-depth experience with GPU programming''. It is correct that we do not use Section 2.2 for automation – our research focus is on identifying simple, but performant programming primitives – not automation.
> >
> > Each part of Section 3 corresponds to terms in Section 2.2:
> > - Section 3.1 addresses $C_{SHARED}$
> > - Section 3.2 addresses $C_{SYNC}$
> > - Section 3.3 addresses $C_{HBM}$, $C_{L2}$, and $C_{SETUP}$
> > This organizational structure is reflected at lines 195-196, 220-221, 238-240, 276-278, 285-287, 359-361, using line numbers from our revised submission.
> >
> > We also added new citations to point to prior work in the computer architecture literature that articulates this cost model to highlight that it is commonly used as a rule of thumb.
> >
> > ## Addressing question 6: TK manages layouts.
> > The review asks ``do users need to manually manage data organization and specify layouts in TK?''. We reiterate that TK manages the layouts for the user (lines 88-94, 275-278 state we automatically choose).
> >
> > ## Addressing question 7: Role of Figure 5
> > The goals of Figure 5 are to (1) show the reader how work is partitioned between load workers and compute workers and (2) show the reader a broader set of TK library functions (warpgroup, TMA). We have clarified the goal of the figure in the caption.
> >
> > ## Addressing question 8: Quantized kernels.
> > Yes TK handles quantized kernels. We include new results for a quantized FP8 GEMM in Appendix B2 and compare to CuBLAS. Our GEMM kernel performance is shown below:
> >
> > **Table: TK FP8 GEMM vs CuBLAS Performance**
> >
> > | Sequence Length | Performance (TFLOPs) |
> > |----------------|---------------------|
> > | 4096           | 1457 vs 1439       |
> > | 6144           | 1507 vs 1509       |
> > | 8192           | 1532 vs 1534       |
> > | 12288          | 1570 vs 1429       |
> > | 16384          | 1447 vs 1396       |
> >
> > ## Addressing question 9: Causal attention kernels.
> > We add new results for causal attention in Section 4. Our kernel performance is shown below:
> >
> > **Attention Performance Comparison (D = 128, B = 16, H = 16)**
> >
> > | Sequence Length | TK Attention Inference Causal (vs FA3) | TK Attention Backwards Causal (vs FA3) |
> > |----------------|---------------------------------------|---------------------------------------|
> > | 768            | 290 TFLOPs (vs 286 TFLOPs)           | 185 TFLOPs (vs 153 TFLOPs)           |
> > | 1536           | 417 TFLOPs (vs 465 TFLOPs)           | 330 TFLOPs (vs 264 TFLOPs)           |
> > | 3072           | 519 TFLOPs (vs 581 TFLOPs)           | 400 TFLOPs (vs 362 TFLOPs)           |
> > | 6144           | 537 TFLOPs (vs 617 TFLOPs)           | 468 TFLOPs (vs 422 TFLOPs)           |
> > | 12288          | 550 TFLOPs (vs 598 TFLOPs)           | 494 TFLOPs (vs 449 TFLOPs)           |
> >
> > ## Addressing question 10: Experimental configurations.
> > We have added Appendix B1 to recap our experimental configurations.
> > - Our PyTorch results are compiled with torch.compile and we have noted this in the paper.
> > - We added the versions of PyTorch, torch compile, and CUDA that we use.
> > - We added the cuBLASt baseline and CUTLASS to our GEMM plots. We use cuBLASLt autotuning for each dimension.
> > - PyTorch vs. Triton:  While both torch-compiled PyTorch and explicit Triton kernels can use Triton as the backend, our PyTorch baselines are written in PyTorch library functions.
> >
> > ## Minor details and presentation
> > We have updated our paper to discuss opportunities for using specialized hardware with TVM and Triton, thank you for pointing this out! Broadly, we are very excited about TVM and Triton and their value to the community. The goal of our work not to promote the use of ThunderKittens over other frameworks. Our work simply explores whether there exists a small set of programming primitives that is sufficient for high performance.
> >
> > Regarding extensibility, indeed, Triton allows for inline PTX. However, the mechanism for this, inline_asm_elementwise, supports a restricted subset of PTX’s capabilities, which represent simple element-wise maps over tensors. Furthermore, while TVM can access tensor cores, there are frequently important low-level optimizations that cannot be expressed within its scope. For example, FlashAttention-3 accomplishes the transpose of the V matrix through two low-level optimizations, one of which perform a partial transpose but also induces a permutation on the elements of the matrix. These sorts of optimizations are not accessible without full and broad PTX support, which is possible in embedded frameworks like CUTLASS and TK, but not in TVM and Triton.
> >
> > Finally, we have addressed the presentation issues -- we added mention of TMA to Section 2, and have fixed the broken links in Appendix B. Thank you very much for the feedback!

---

> > > ### Comment · Reviewer_ANEJ · 2024-11-22
> > >
> > > Thanks for the detailed response! I truly appreciate the authors’ efforts to conduct additional experiments, clarify misunderstandings, and enhance the paper’s presentation. I will accordingly raise my score. A few additional suggestions:
> > > 1. Please incorporate the Feature-TK-CUTLASS-Triton comparison table into the paper. This table is both important and valuable for helping readers understand the differences among these frameworks.
> > > 2. Please include the discussion on “highlighting TK’s value to the ML audience” in the paper, which effectively explains why the ML audience should care about this work.
> > > 3. In Figure 7, the text for the CuBLASLt-2048 case overlaps with the legend. Please adjust for clarity.
> > > 4. Regarding the mention of “an example ThunderKittens ping-pong GEMM kernel in the appendix,” is this referring to Figure 15? If so, please explicitly clarify this in the text.
> > > 5. In Appendix B.1, please provide a specific commit ID for the Triton compiler. As Triton evolves rapidly, the performance of different commits—even under the same version number (currently 3.0.0)—can vary significantly. Including this information would ensure better reproducibility.

---

> > > > ### Author Response · Authors · 2024-11-23
> > > > **Thank you!**
> > > >
> > > > Thank you again for your review and help in improving the paper! We appreciate the score update and will be sure to fix these additional points.

---

### Official Review · Reviewer_nd8y · 2024-11-01

**Soundness:** 3
**Presentation:** 2
**Contribution:** 3
**Rating:** 8
**Confidence:** 4

**Summary:**

This paper proposes a framework to facilitate easy writing of efficient CUDA kernels. The authors leverage the asynchronous compute capabilities of the Hopper series GPUs by following a producer-consumer paradigm, to efficiently overlap different kernel operations. Additionally, the authors investigate the impact of various memory ordering strategies, demonstrating that relatively simple strided patterns offer the best tradeoffs. Lastly, the authors demonstrate performance that is comparable to or exceeds existing methods, including Triton.

Overall, the work provides a significant contribution to improving computational efficiency for common operations, though the application appears limited in scope. Additionally, minor technical and structural errors impact readability. These issues could be addressed in a revision, at which point I would be inclined to raise my score.

**Strengths:**

The authors demonstrate significant improvements to computational efficiency within a clearly defined framework that appears relatively straightforward to adapt. Their framework also provides functionality for more complex resource management, which is often challenging to manage directly in CUDA.  Additionally, the authors demonstrate the impact of varying hyperparameters for several key kernel operations, most of which match or exceed standard baselines. Lastly, the results show a surprising contrast with Triton implementations, positioning their approach within the CUDA domain while achieving a similar level of complexity to Triton.

**Weaknesses:**

-	The application appears limited in scope, which should be explicitly addressed. For example, is the framework limited to Hopper GPUs and above? And the focus on 16x16 register blocks may limit extensibility to other common cases such as GEMV and sparse computations.
-	The paper contains many issues with presentation, including caption errors, grammatical and awkward wording, and typos, all of which impair readability.
-	The paper overlooks relevant computer architecture literature regarding performance modeling, specifically in the context of balancing compute and memory (e.g. roofline analysis). Many of the findings presented in the paper are expected from the existing literature.

**Questions:**

1)	Is your framework limited to the Hopper series? Can it be applied to A100s, or other GPUs such as the A40/L40?
2)	You focus on the 16x16 register block level, but how can your framework be extended to smaller blocks, such as with GEMV, sparse operations, and masked operations (e.g. non-power-of-two dimensions and strided masking, such as in Natten).
3)	Throughout the paper, you focus on BF16 precision (with the exception of softmax); have you considered other data types, such as integer types or floating-point formats like FP8?
4)	How could your framework be extended to handle multi-GPU operations, such as Fully Sharded Data Parallel (FSDP) for split operations? This seems like a natural extension of the producer-consumer model.
5)	You compare yourself against Triton, which also supports AMD GPUs. Can you address this as a potential tradeoff in the paper? Alternatively, if your framework can be trivially extended to ROCm, this should be included in the paper with a demonstration, otherwise it represents a tradeoff between efficiency and portability.
6)	Your cost model in Section 2.2 is effectively a Roofline model; could you contextualize this in the existing literature? The results in Table 3 are expected, as reordering increases the arithmetic intensity (FLOPs/Byte) of the inner loops.
7)	Throughout the paper, the emphasis on industry versus academic adoption (including the use by undergraduates) feels extraneous and detracts from the main narrative. The paper’s contributions should stand on their own without reliance on external endorsements or applications.
8)	Figures 2 and 5 present a simplified sketch for softmax, whereas the true implementation is significantly more complex, potentially leading to a misleading comparison with PyTorch.  Furthermore, Figure 2 led me to question why you are using C at all for the API, when the listing could easily have been captured by a python trace (e.g. Triton). This design choice is only clarified upon reviewing the implementation details provided in the appendix and supplementary material.

To build on these questions, the feedback below addresses specific technical details and aims to enhance overall clarity. While this paper presents a strong contribution toward improving kernel efficiency, addressing these points will better showcase the authors’ contributions.

Minor Technical Errors:

-	044: The H100 datasheet shows a 7.4x ratio between TCs and ALUs, not 16x. Additionally, my understanding is that the TCs necessarily require bubbles as the Register path cannot keep up with the TC I/O for full throughput.
-	136: This should be "can load" or "may load" instead of "loads." In general, a kernel does not necessarily need to load data from memory. Kernels can rely solely on arguments (loaded into registers at startup) to generate new data. For example, a kernel might generate a pseudo-random noise tensor without accessing memory.
-	148: The 32 threads must be within the same quadrant, where “consecutive” or “adjacent” would be more appropriate than “nearby”.
-	150: In Ampere, a warp cannot simultaneously occupy different functional units, though separate warps can. For accuracy, please verify this claim against the Hopper documentation or micro-benchmarking paper, otherwise consider omitting if verification is unavailable.
-	167: Excess registers spill over into Global Memory, not L1. They can appear in L1 due to the memory hierarchy, but this is at the discretion of the hardware cache manager.
-	171: Multiple thread blocks can only schedule on the same SM if there is sufficient space (e.g. SMem), otherwise they would clobber each other.
-	173: This statement should be more precise to mention “all thread blocks” and that the L2 is hardware managed, making it distinct from the software managed SMem.
-	179: The tail-effect cost mentioned only applies to singular kernels. Ideally the GPU should have multiple kernels in flight, which can run concurrently.
-	It would also be relevant to mention that kernels which contain too many instructions can cause slowdown as they will incur ICache misses.

Presentation Issues:

-	The abstract should be revised for clarity, with suggested improvements like “creates a”, “suggest that”, and “resembling PyTorch.”
-	The paper could benefit from clarity revisions in several sections, where phrasing and word choice could make technical details easier to follow. Lines: 073, 170, 178, 205, 278, 299, 301, 328, 370, 397
-	325: You should not use "[1]" and "[2]" to enumerate concepts as they are easily confused with reference indicators.
-	Table 2 and Table 3 should probably be Figures like Figure 6. It is also unclear why these stop at 4 stages, K=1024, and what K is. (MxN)x(NxK)?
-	Figure 7 and 8 should use subfig captions rather than plot titles. If parameters are common among subfigures, then they should be stated in the figure caption, otherwise in the subfig caption. The fontsize for the axis and labels is too small. Finally, the batch size does not match with the titles and caption.
-	The table in Section 4.2 is missing a caption and column (TK is listed twice).
-	The reference links are broken in Appendix B.

---

> ### Author Response · Authors · 2024-11-22
> **Response to review**
>
> Thank you for your review, we are very grateful that you took the time to provide the thoughtful detailed feedback, which has helped us significantly strengthen our work. We carefully address your feedback in our response and hope that it helps.
>
> ## Demonstrating the broad scope of ThunderKittens
> Thank you for the question! We provide new results beyond the submission for FP8 GEMM, causal attention, NVIDIA 4090 kernels, and Apple M2 kernels to further emphasize the scope of our framework. We also show that TK supports tile dimensions that are not multiples of 16; we provide demo attention kernels that use this feature. Please find these results in the common response and in Appendix B of our revision. We also added new discussion of our implementations for these features in Appendix D.1.
>
> ## Connections to the computer architecture literature
> Thank you for raising this concern! We have contextualized both the cost model in Section 2.2 and the results in Table 3 to highlight the existing literature. We did not aim for Section 2.2 to appear as our contribution and appreciate the suggestions to further establish that this is fundamental background knowledge. We correspondingly added citations to computer architecture works in this section. Please let us know if any other changes are helpful!
>
> ## Extending to Fully Shared Data Parallel (FSDP)
> We appreciate this suggestion. While multi-GPU operations like FSDP are indeed important for large-scale training, our current focus is on optimizing single-GPU kernel performance. Users of our framework can currently handle multi-GPU communication through established libraries like NCCL, which provides highly optimized primitives for inter-GPU collective operations. We agree that extending the producer-consumer model to multi-GPU contexts is an interesting direction. We are excited to explore such directions in future work.
>
> ## Presentation improvements
> We have made several presentation improvements to improve readability. We apologize for the presentation errors. Additionally:
> - We removed the mention of external endorsements from the paper, per your recommendation.
>
> - The review notes that Figures 2 and 5 are too simplified compared to the full implementation. We have updated the paper to include more precise captions, to highlight that the figures represent specific aspects of the library: Figure 2 is only meant to show that the library functions and tiles are Pytorch-like. Figure 5 is only meant to improve intuition on separation of responsibility across workers.
>
> Thank you for your feedback on presentation, which has helped improve the paper.
>
> ## Technical details
> We have updated the paper to reflect your feedback, and appreciate your help in tightening the technical writing. We also provide specific responses below:
> - With respect to the TC:ALU ratio: We were specifically referring to the BF16 tensor core to ALU compute ratio, while your citation refers to FP16 FMA performance. Nonetheless, there was a slight error in our writing for the H100; we've adjusted the number to match NVIDIA's official 15x figure. (The A100 GPU officially has 16x.)
> - Regarding kernel memory access: Kernels can operate without memory access. We’ve updated the wording to "typically loads" for accuracy, as it describes the typical pattern for deep learning kernels, which is our focus.
> - On warp execution units: We've revised this section to be more precise. You're correct that individual warps cannot simultaneously occupy different functional units. We've clarified that we were referring to instruction-level parallelism (e.g., overlapping memory loads with computation) and concurrent execution across different warps.
> - Concerning register spills: We've updated this to be more precise. Register spills occur to local memory, which may be cached in  L1 depending on hardware policies. Thank you for helping us make this clearer.
> - On SM thread block scheduling: We've added explicit mention of resource constraints as a factor in thread block collocation.
> - As to kernel tail effects: While concurrent kernel execution is possible across multiple asynchronous CUDA streams, our discussion focuses on the common case in deep learning where kernels execute sequentially within a single CUDA stream, and for these workloads tail effects are an important consideration. We've added a note acknowledging advanced multi-stream techniques while maintaining our emphasis on typical AI workloads. We are excited to explore overlapping kernel execution in future work, such as in the recently-released async TP workload.
> - We've also added a brief note about instruction cache considerations, though this wasn't a significant factor in our evaluated kernels.
>
> Please let us know if anything else would be helpful in your review!

---

> > ### Author Response · Authors · 2024-11-25
> > **Following up**
> >
> > Dear Reviewer nd8y,
> >
> > Thank you again for your review! We are wondering if our response has addressed your concerns? Please let us know if we can provide anything more and we look forward to your response!

---

> > > ### Comment · Reviewer_nd8y · 2024-11-27
> > > **Response to Authors (1/2)**
> > >
> > > I thank the authors for their response and commend their efforts in extending TK support to both the 4090 and Apple M2.
> > >
> > > ---
> > >
> > > ## Demonstrating Broader Scope
> > >
> > > Thank you for including these cases; the competitive performance across the added conditions is impressive and further supports the generality of your method. However, the new results raise a few questions.
> > >
> > > 1. Flash-Attention2 appears to perform better on the 4090. Have you identified the cause of this? While pinpointing the reason would be helpful, I consider this a secondary concern, as the advantages of a consistent framework and greater adaptability can outweigh minor performance variations.
> > >
> > > 2. Similarly, Figure 8 indicates that FA3 outperforms TK in the causal attention forward pass. Have you identified the reason for this? Could it be related to the use of 16x16 blocks?
> > >
> > > 3. You explored adaptation to the 4090, which is an Ada GPU incorporating many architectural changes introduced by Hopper. Is this focus due to available testing hardware, or is there a fundamental restriction preventing testing on Ampere? While Ampere is no longer state-of-the-art, it remains widely used in academic clusters (e.g., A100, A6000) and among consumers (e.g., 30 series).
> > >
> > > 4. Thank you for discussing block padding with masks. However, this does not fully address the concern with GEMV. While GEMV is rarely used during training, it is the primary operation for non-batched LLM inference. As it stands, your framework implies GEMV would incur a 16x performance reduction due to padding. Could you clarify or address this concern?
> > >
> > > ---
> > >
> > > ## Connections to Computer Architecture Literature
> > >
> > > Thank you for adding the reference to the Roofline model in Section 2.2. However, the wording in this paragraph may still need refinement to avoid implying that your cost model is novel. Perhaps it could be described as a “reformulation” of the Roofline model rather than inspired by it. Notably, roofline considers FLOPs, which is equivalent to $1/\mathrm{cost} \sim 1/\mathrm{time}$ in your model.
> > >
> > > Regarding the contextualization of Table 3, I did not find this addressed in your revision. I recommend framing the reordering as increasing the arithmetic intensity (FLOPs/byte), which aligns with the insights provided by the Roofline framework. This should suffice to contextualize the result in Table 3. Currently, the text may be interpreted as offering a fundamental new insight.
> > >
> > > ---
> > >
> > > ## Extension to FDSP
> > >
> > > Thank you for your response. Perhaps this could be included in the Camera-Ready version (in the appendix). Alternatively, you could provide an example in your code release that demonstrates computing GEMM across two GPUs with TK using PyTorch as an intermediary.
> > >
> > > Notably, this extension is important not only for training but also for inference. Many LLM frameworks, such as vLLM, support tensor-parallel computations to efficiently distribute weights and increase parallelism.
> > >
> > > ---
> > >
> > > ## Presentation Improvements
> > >
> > > Thank you for addressing the suggested changes. I noticed that some issues highlighted in my original review remain unresolved. However, these remaining issues appear to have only a minor impact on overall readability.

---

> > > > ### Author Response · Authors · 2024-11-29
> > > > **Response to review (1/2)**
> > > >
> > > > Thank you very much for your response! We’ve done our best to address your questions below, and will also take them into full account for our next revision.
> > > >
> > > > ### Demonstrating Broader Scope
> > > >
> > > > We have addressed your comments on scope below. We are grateful that you appreciated the advantage of a consistent framework, including: TK is running on multiple AI workloads and hardware platforms, competing with the best baselines on each of those settings--with the same abstractions. Each of TK’s abstractions can of course be further optimized if the need arises – as a C++ embedded framework, the user can incorporate the full power of CUDA when using TK. However, we demonstrate that with few abstractions, TK already outperforms major projects from prior work – there are full research projects for each individual baseline kernel we compare to (e.g., flash attention).
> > > >
> > > > **FA2 Performance**: As you note, we do find about a 5% performance difference on a 4090 relative to FlashAttention-2. We do know the source of the issue, which has to do with shared memory. In the case of the 4090 kernel, which does not have the benefit of TMA acceleration to compute addresses and load data into tensor cores, the kernel consumes additional resources computing these shared memory addresses. We do not believe this is a fundamental limitation and can address this for the camera ready. Specifically, the additional cost we incur could be amortized through TK internally caching certain address offsets in a few extra registers; we were not able to implement this in-time for rebuttals. But in any case, we are excited that our naïve port of the TK primitives transfers with relatively minor performance penalty, even within the rebuttal timeframe.
> > > >
> > > > **FA3 Performance**: In the case of FA3, the reason is actually not related to 16x16 blocks, but instead an algorithmic difference. The kernel we implement in ThunderKittens is not FA3's algorithm, but rather a modified version of FA2, updated for the H100 GPU. In our causal forward attention kernel pass, we therefore pay an additional cost from the fact that our kernel launches three consumer warpgroups instead of two, which means more threads sit idle at the end of the kernel. We do this in order to show that most of the performance improvements of FA3 actually come from a relatively straightforward update into the Hopper architecture, without complex ping-pong scheduling. This is core to our research thesis that a small number of primitives can go a long way for AI kernels.
> > > >
> > > > **4090 and A100**: We focused on adaptation to the 4090 actually because (a) we have 4090 GPUs more easily available to us, and (b) we felt it was more different from the H100 in terms of performance characteristics. We are happy to include A100 for the camera ready. We hypothesize that the A100 will face a similar address generation cost as the 4090, though as we wrote above, we do believe this gap could be closed without breaking any tile abstractions or even adjusting any memory layouts.
> > > >
> > > > **GEMV**: Regarding GEMV, we have not yet optimized its performance, as even in non-batched LLM inference, techniques such as speculative decoding are now ubiquitous in production scenarios. And indeed: the 16x cost penalty of padding is about the same as one incurs simply by not using the tensor cores, so one might as well put something in those columns. (Furthermore: it is probably still actually preferable to use the tensor cores in most such scenarios in order to save instruction issue slots, so that other pipelines can be used in parallel.) But there is no reason whatsoever that our framework could not support an ALU-based GEMV; it would just be another primitive like mma or row_max or whatnot.
> > > >
> > > > ### Connections to Computer Architecture
> > > > We concur that our cost model is not novel, and we have zero interest in claiming it as a contribution of the work. We don’t think it’s all that interesting, so we’d actually be pretty sad if anyone thought of it as a contribution. But we do believe it’s important for a non-systems audience to provide a broad framing of how one might think of cost on the GPU. We will certainly further emphasize this in a camera ready (e.g., removing “we show”, changing “inspired by”).
> > > >
> > > > Your idea of adding a third column to Table 3 with the empirical arithmetic intensity is a good one -- apologies we didn’t understand it last time around. We agree it would help connect to the original model, and we’ll be sure to update that.
> > > >
> > > > ### FSDP
> > > >
> > > > Respectfully, we believe this request is beyond the scope of our work. Gluing two TK kernels together with Pytorch would not really serve to add to the contributions of the work, in forming a basis for writing individual kernels, and in fact would distract from the main body of the work. We certainly concur that tensor-parallel is an important regime for many, many workloads, but in our view networking is best treated as a meaningfully distinct problem from kernel development.

---

> > > > > ### Author Response · Authors · 2024-11-29
> > > > > **Response to review (2/2)**
> > > > >
> > > > > ### Technical Details
> > > > >
> > > > > The TC:ALU ratio comes from both the datasheet as well as our own microbenchmarks. Regarding the datasheet, https://resources.nvidia.com/en-us-tensor-core/nvidia-tensor-core-gpu-datasheet, affirms that for an H100 SXM, there are 1979 TFLOPs of sparse BF16 TC compute, which corresponds to 989 TFLOPs of dense BF16 compute (see more here https://developer.nvidia.com/blog/structured-sparsity-in-the-nvidia-ampere-architecture-and-applications-in-search-engines/).  Additionally, there are 67 TFLOPs of FP32 ALU compute. Thus the ratio is 989/67=14.8x. Additionally, here is a simple benchmark to affirm that BF16 does not run at double-rate relative to FP32, even though NVIDIA’s Hopper Architecture In Depth documentation (https://developer.nvidia.com/blog/nvidia-hopper-architecture-in-depth/) suggests that it would (with the caveat that NVIDIA admits those numbers are preliminary): https://drive.google.com/file/d/1QaUgHjTpt9Ljb4_3yoniZ6AOx0p8kQKC/view?usp=sharing
> > > > >
> > > > > Regarding the discussion of register spilling: first, we apologize, we thought we had updated that text to mention further spills up the cache hierarchy (though, as you note, this only understates our point more modestly) -- it must have gotten lost somewhere. We’ll be sure to put that back in. However, it’s worth noting that it’s also somewhat inaccurate to describe as spilling to global memory, since the PTX model is explicit that local address space is distinct from global address space. Part of the reason for our elision in this discussion was that we felt that a full treatment of the half-dozen address spaces present in NVIDIA’s model was well beyond an appropriate introduction of GPUs to an AI audience.
> > > > >
> > > > > Believe it or not, the operations of min/max and FMA are not computed on the same units. On an H100, FMA runs through the FMA pipeline (which is actually composed of two sub-pipelines, FMA heavy and FMA lite), but floating point min/max are actually computed on the ALU pipeline. This can be seen with a simple dummy kernel that runs only min/max or only FMA’s, and profiling in ncu. It’s further confirmed by NVIDIA’s kernel profiling guide, https://docs.nvidia.com/nsight-compute/ProfilingGuide/index.html. We bring this up in the paper specifically because it can be counterintuitive, but understanding which operations can and cannot be overlapped can meaningfully impact kernel performance.
> > > > >
> > > > > Throughput differing across pipelines is indeed accurate; this is not mere resource contention, and operations do not all run at one operation per clock per data lane. The clearest documentation of this can be found within NVIDIA’s CUDA C++ Programming Guide at https://docs.nvidia.com/cuda/cuda-c-programming-guide/#arithmetic-instructions. To illustrate: from that chart, one can see that FP32 FMA runs at 8x the rate of FP32 transcendental functions (e.g. expf), and as an FMA is generally counted as two operations, it runs at a full 16x the FLOPs. This is a particularly extreme example, and it aligns with our own microbenchmarks.
> > > > >
> > > > > Fixed instruction latency is actually a distinct modality of thread stall from resource contention. Generally speaking, resource contention is observed as a “pipe throttle” stall of some kind, depending on the operation. For example, contention on load units is observed as a “load global throttle”, contention on transcendental functions usually appears as an “MIO throttle”, and contention on ALU resources appears as a “math pipe throttle”. In contrast, a fixed instruction latency (for example, an FMA needs to return a result to a register before it can be used in a store to global memory) is observed as a “stall wait:”. Different workloads can consume identical resources and yet have different performances, depending on to what degree instruction-level parallelism allows threads to issue new instructions before previous instructions have finished.
> > > > >
> > > > > Regarding the instruction cache: we added a brief discussion of it, per your request. However, although it is indeed possible to construct kernels which suffer from terrible instruction cache misses, we have not found this to be a significant effect in any of our kernels, whereas register spilling is a frequent, important, and tricky problem to address. Accordingly, we do feel it’s important to maintain the focus of the background onto the key problems we’ve observed.

---

> > > > > > ### Comment · Reviewer_nd8y · 2024-11-30
> > > > > > **Response to Authors**
> > > > > >
> > > > > > Thank you for addressing my additional questions and investigating the performance discrepancies with FA2 and FA3. I also appreciate the clarification that TK is applicable to the older Ampere generation.
> > > > > >
> > > > > > ---
> > > > > >
> > > > > > **GEMV:** Regarding GEMV, are there any current implementations successfully utilizing 16x speculative decoding? My understanding is that 4x is standard, which could still result in a significant performance penalty. However, I had not considered that padding might still be more efficient. If true, this would indeed be an interesting insight.
> > > > > >
> > > > > > **TC:ALU Ratio:** Comparing BF16 TC FLOPs to FP32 FMA FLOPs seems somewhat misleading. A more appropriate comparison would be against BF16 FMA FLOPs for a fair evaluation.
> > > > > >
> > > > > > **Register Spilling:** I believe that simply connecting to the cache hierarchy (DRAM/HBM) is sufficient. Discussing the various memory regions would introduce unnecessary complexity.
> > > > > >
> > > > > > **ALUs vs FMA:** Thank you for the clarification; this is both interesting and surprising. It is also consistent with the throughput table. In hindsight, this makes sense as it unifies integer and float min/max operations without requiring explicit FSUB support.
> > > > > >
> > > > > > **Throughput:** The throughput tables appear to confirm my original statement. The example of expf is attributable to the XU count: 16 per SM versus 128 FMA units per SM. While each unit maintains a throughput of 1/clk, XU instructions must be partially serialized within the warp.
> > > > > >
> > > > > > **Instruction Latency:** Thank you for clarifying the observed cases. While I agree that FMA has a fixed latency, the 'stall wait' described does not result from this latency alone. It arises due to data hazards (RAW dependencies) in low-ILP scenarios, where dependent instructions must wait for the result. In contrast, in high-ILP workloads with independent instructions, the same fixed latency would not cause a stall. This differs from what is typically considered 'fixed instruction latency,' such as a division operation requiring N cycles to complete and exhibiting a 1/N throughput. In such cases, even independent division instructions must wait the full latency. However, I believe this distinction may introduce unnecessary complexity, which is why I suggested avoiding the issue by listing 'resource contention' instead.
> > > > > >
> > > > > > ---
> > > > > >
> > > > > > I thank the authors again for their detailed responses and for addressing the majority of my concerns. I am satisfied with their efforts and will raise my score accordingly.

---

> ### Comment · Reviewer_nd8y · 2024-11-27
> **Response to Authors (2/2)**
>
> ## Technical Details
>
> Thank you for addressing these corrections. However, a few issues remain:
>
> 5. Regarding the TC:ALU ratio, where are you finding this number? According to the H100 datasheet, NVIDIA reports a 7.4x increase, not 16x. This is the case for both FP16 and BF16, as they utilize the same hardware units, likely to support TF32. Are you perhaps considering the FLOPs with sparse matmul?
>
> 6. The discussion of register spilling to L1 still seems imprecise. It would be more accurate to describe this as “spills into ~~global memory~~ HBM.” The subsequent sentence effectively explains how SMEM and L1 can be traded to provide additional L1 capacity for these spilled registers, where I believe the local memory L1 policy is write-back but evictions can occur during global memory load / stores.  However, it is worth implying that, in the most general case, spilling registers may incur a full hierarchy cache miss, potentially causing significant performance degradation. As currently written, this performance issue may be understated.
>
> 7. There are a few minor technical inaccuracies in Section 2.1 that I’d like to highlight:
>     - The ALU operations of min/max and FMA are computed on the same units, but the text seems to suggest they are distinct. While speculative, these operations likely share significant overlap within the datapaths of the ALU/CUDA Core.
>     - Could you confirm whether the statement “throughput differs across pipelines” is accurate? Or is the observed throughput difference due to resource contention rather than variations in individual unit throughput? My understanding is that all compute units, including the XFU, have a consistent throughput of 1 op/clk.
>     - Regarding thread stalling, the term “fixed instruction latencies” may not be sufficiently precise. I suggest replacing it with “resource contention,” which provides a broader and more accurate explanation.
>
> Additional Comments
>
> - Regarding the instruction cache, this is particularly relevant to your discussion on reducing the number of registers per thread. An increase in operations could lead to more instructions, potentially resulting in I$ misses.
> - Thank you for your response regarding kernel tail effects. I had not considered the limitation of sequential kernel launches, and this would indeed be an excellent direction for follow-up TK!

---

### Official Review · Reviewer_kgD5 · 2024-11-04

**Soundness:** 3
**Presentation:** 3
**Contribution:** 3
**Rating:** 8
**Confidence:** 4

**Summary:**

The paper proposes a new programming library for implementing efficient CUDA kernels. tThe paper contains the three ideas at three different levels for CUDA kernel implementation: (1) At warp-level, the author proposes to organize the tile as the multiple of 16; (2) at thread-block level, the author devises template libraries to overlap between different asynchronous warps; (3) at grid-level, the author proposes methods for managing kernel launch kernel launching overheads. As a result, the proposed library can achieve a performance on par with the existing state-of-the-art implementations.

**Strengths:**

* The paper proposes methods at different levels that simplify CUDA kernel implementations
* The paper can achieve a similar performance compared to the state-of-the-art implementation

**Weaknesses:**

* The paper has not discussed the tunning overhead with the proposed techniques.

**Questions:**

Thanks for submitting the excellent paper to ICLR. While in general I enjoyed reading the paper, I have a few thoughts on the extension of the paper. Specifically, this paper proposes a new CUDA abstraction that allows users to write new kernels. However, it seems that it is built on top of the fact that all the dimensions should be a multiple of 16. This could be problematic in the context of dynamic shapes where the dimension does not divide 16. Could you please elaborate on how could the proposed technique be extended to such cases?

Besides, the paper uses auto-tuning for further adjust the hyperparameters for a better performance. Could you elaborate how much the tunning overhead is?

---

> ### Author Response · Authors · 2024-11-22
> **Response to review**
>
> Thank you for your positive and thoughtful review! We are glad that you appreciate TK’s simplicity and performance, including providing kernels that far outperform the prior state-of-the-art kernels by up to 14x. Here, we address your remaining questions.
>
> ## Simplicity of tuning kernels within TK
>
> In our work, we de-emphasize autotuning, as we find that tuning kernels is usually straightforward once the right primitives are in place. This is, in a sense, a limitation of our framework compared to compiler-based frameworks like Triton, which automatically block and tile. However, our core argument is that for most AI workloads, sophisticated ML compilers are not needed to concisely, performantly, and portably express kernels.
>
> To the extent that tuning is necessary, it can usually be done by twiddling occupancy and pipeline depth, for which there are usually only a few values that make sense, anyways. **TK makes it easy to twiddle these two values** by changing a single number in the kernel, and recompiling.
>
> For tile size optimizations, users may need to consider hardware constraints such as shared memory and register limitations, requiring more careful analysis. However, **TK's templated operation design minimizes the code changes needed to implement these adjustments**.
>
>
> ## TK supports dynamic shapes beyond multiples of 16
>
> Thank you for the insightful question with respect to the potential limitations of 16x16 tiles. Yes, TK does support workloads where the dimensions are not multiples of 16. To address your feedback, we include new results in Appendix B.2, where we report that TK gives high performance on attention kernels that require tiles that are not multiples of 16.
>
> We also describe how we support these dimensions in Appendix D.1. The common solution to support any dimension is to use *padding*.  ThunderKittens supports padding within the kernel, and other frameworks, such as CUTLASS and Triton also pad data into tiles. Padding is the solution the hardware prefers, too, since tensor cores generally operate in multiples of some base size (usually 16).
>
>
> We hope these responses help address your feedback, and please let us know if you have any additional questions!

---

> > ### Comment · Reviewer_kgD5 · 2024-11-26
> >
> > Thanks the authors for the response. I have read it and it have addressed all my concerns.

---

### Author Response · Authors · 2024-11-22
**Common response to all reviewers**

We thank all the reviewers for their time and effort reviewing our work and for their constructive comments, which have made our paper stronger. Reviewers consistently appreciated our “compelling”, “impressive” and “significant” performance improvements compared to the prior state of the art kernels across numerous AI workloads [kgD5, nd8y, ANEJ, 5GXe], and the “simplicity”, “freshness”, and “usefulness” of the ThunderKittens (TK) programming framework [kgD5, nd8y, ANEJ, 5GXe]. Reviewers also appreciate that our work can help make hardware-aware AI accessible to a broader set of developers [ANEJ, 5GXe].

In this common response, we provide: (1) a recap of our contributions and the changes we made in our revisions, and (2) details on important new results relevant to all reviewers. Please find our comments for individual reviewers in their respective threads.

## Summary of contributions
We explore whether extracting peak performance from AI hardware requires complex kernels, or whether simple programming primitives that capture most of the optimization patterns needed in AI. We are inspired by widely used programming frameworks: CUTLASS, which supports every hardware optimization edge case through layers of templating, and Triton, which exposes fewer hardware optimizations, but provides a drastically simpler programming experience: Our research takes initial steps to explore the large tradeoff space between programming complexity and the accessibility of peak hardware performance. Our contributions are:

1. **Programming primitives**: At each level of GPU parallelism, we identify opportunities to simplify the programming experience, without sacrificing on the accessibility of peak performance. To help ML researchers use these opportunities, we release the ThunderKittens (TK) programming library. Our optimizations include:
- **Warp-level**: We automatically select and manage memory layouts for the user. We provide optimized library functions inspired by PyTorch (mma, exp, cumsum).
- **Block-level**: We provide a kernel template that helps users utilize the core asynchronous execution patterns that are common in AI workloads.
- **Grid-level**: We provide persistent grid support and a template function to help users manage L2 cache reuse for the kernel.

2. **Optimized kernels**: We evaluate using a **broad range of popular AI workloads** (attention variants, convolutions, state space models, GEMM, positional encodings, layernorms), **4 data types** (BF16, FP16, FP32, FP8), **3 different AI hardware platforms** (NVIDIA 4090, H100, and Apple M2). We compare kernels written in TK to the strongest baselines available: state-of-the-art, carefully engineered kernels written in frameworks – CUTLASS, Triton – that are supported by large industry teams over multiple years. Excitingly, we find that TK kernels consistently compete with state-of-the-art, despite using **<200 lines of code per kernel** on average.



## Common results and responses
Here we provide results and responses to two points that were of interest to multiple reviewers:
Breadth of features: Reviewers asked about the breadth of features supported in ThunderKittens and wanted to understand the performance on edge cases.
Comparison to other frameworks: Reviewers wanted to further understand how TK compares to popular frameworks like CUTLASS and Triton.

### Breadth of features supported in ThunderKittens:
Reviewers were curious about whether few edge cases are supported in ThunderKittens. We show that they are indeed supported and add discussion to explain the implementations:
1. **Precision levels:** The review mentions that our paper focuses on BF16 and it would be useful to highlight other data types. Our previously demonstrated kernels rely on FP16, FP32, and BF16. We now provide an **FP8 GEMM kernel**, which achieves up to 1570 TFLOPS on NVIDIA H100s. Please find the results for the FP8 TK kernel in Appendix B2.
2. **Padded tiles:** Reviews mention that the 16x16 tiles may be restrictive for applications requiring dimensions offsetted from multiples of 16. We now also provide results for a TK attention kernel that does not assume multiples of 16. Please find a description of the TK unaligned attention kernel in Appendix B2. We also add Appendix D1 to our revised submission, which details how **TK manages tiles, when dimensions are offset from multiples of 16**.
3. **Range of hardware platforms:** Our original evaluations focused on the top-of-line data center GPU, NVIDIA H100s. **We now provide new evaluations for the top-of-line consumer GPU, NVIDIA 4090s, and personal hardware, Apple M2 chips.** We entirely converted TK to Apple hardware in the span of the rebuttal period and are including this in our revised supplementary materials. Please find performance benchmarks for our new TK kernels in Appendix B. TK consistently provides high performance and is extensible.

---

> ### Author Response · Authors · 2024-11-22
> **Common response to all reviewers [continued]**
>
> # Performance Benchmarks
>
> Please also find these results discussed in Appendix B of our revised submission.
>
> ## TK extends across hardware platforms
> To further highlight the extensibility of TK, we provide new kernels for the NVIDIA 4090 and Apple M2 platforms.
>
> **Table: 4090 Attention FWD (non-causal, batch=16, heads=16)**
>
> | Sequence Length | TK Attention FWD (head dim 64) | TK Attention FWD (head dim 128) |
> |----------------|-------------------------------------------------------|--------------------------------------------------------|
> | 1024           | 150 TFLOPs                                            | 141 TFLOPs                                              |
> | 2048           | 154 TFLOPs                                            | 145 TFLOPs                                              |
> | 4096           | 157 TFLOPs                                            | 156 TFLOPs                                              |
> | 8192           | 160 TFLOPs                                            | 148 TFLOPs                                              |
>
> **Table: Apple M2 Pro Attention FWD vs Apple MLX  (non-causal, batch=16, heads=16)**
>
> | Sequence Length | TK Attention FWD (head dim 64) | TK Attention FWD (head dim 128) |
> |----------------|--------------------------------|--------------------------------|
> | 512            | 3523.5 vs 3088.4 GFLOPS      | 3263.38 vs 3770.5 GFLOPS      |
> | 1024           | 3723.8 vs 3276.9 GFLOPS      | 3435.89 vs 3977.2 GFLOPS      |
> | 1536           | 3761.8 vs 3313.2 GFLOPS      | 3490.66 vs 4053.4 GFLOPS      |
> | 2048           | 3784.1 vs 3309.6 GFLOPS      | 3488.09 vs 4006.0 GFLOPS      |
> | 2560           | 3793.4 vs 3329.8 GFLOPS      | 3483.83 vs 4047.9 GFLOPS      |
>
> **Table: Apple M2 Pro GEMM TK vs Apple MLX Performance**
>
> | Sequence Length | Performance (GFLOPS)           |
> |----------------|--------------------------------|
> | 1024           | 3830.8 vs 3444.7            |
> | 2048           | 5238.8 vs 4839.5            |
> | 4096           | 5600.6 vs 5190.1            |
> | 8192           | 5675.8 vs 5182.7            |
> | 16384          | 5267.0 vs 5117.7            |
>
> ## Comparing TK to CUTLASS and Triton:
>
> We focus on a research question: *What are the tradeoffs between programming complexity and the accessibility of peak hardware performance?* Our novelty is in showing a **new point on the tradeoff space exists**. We can use fewer primitives to achieve higher performance on AI workloads, compared to what’s been demonstrated in prior frameworks.
>
> **Overview:** CUTLASS provides many layers of templated C++ primitives. Compiler approaches like Triton expose fewer optimizations, but simplify the programming experience. Researchers often turn to Triton, pointing to the relative simplicity of Triton. Our novelty is showing a small set of C++ primitives that captures most of what we need and and consistently unlocks state-of-the-art kernels across hardware platforms. Identifying this small set is not trivial given the complexity of GPUs.
>
> Below, we summarize how Triton, CUTLASS, and TK support key features needed for high performance:
>
> **Table: Framework Feature Comparison**
>
> | Feature                                 | TK     | CUTLASS | Triton |
> |----------------------------------------|--------|---------|---------|
> | Direct register memory control         | YES    | YES     | NO      |
> | Fine-grained asynchronous execution    | YES    | YES     | NO      |
> | PyTorch-like library functions for usability | YES    | NO      | YES     |
> | Supports multiple hardware platforms    | YES    | NO      | YES     |
>
> As proxies for the simplicity versus efficiency resulting from different frameworks, we measure: (1) the size of various frameworks and (2) the lines of code used across kernels. We validate two claims:
> 1. TK compares to Triton in lines of code, and both outperform CUTLASS
> 2. TK competes with or outperforms Triton and CUTLASS/raw CUDA baselines
>
> **Table: Library Size Comparison**
>
> | Library  | Size (Bytes) | Date/Version |
> |----------|-------------|--------------|
> | CUTLASS  | 22 MB       | 10/22/2024   |
> | Triton   | 12.6 MB     | 10/22/2024   |
> | TK       | < 1.0 MB    | 10/22/2024   |
>
>
> **Table: Lines of code across TK H100 kernels, state of the art non TK kernels, and the TK speed up over the reference kernel.**
>
> | Workload | TK kernel (lines of code) | Reference kernel (lines of code) | Speed up (max-min) |
> |----------|--------------------------|--------------------------------|-------------------|
> | Attention forwards | 217 | 2325 (CUTLASS FA3) | 0.87-1.14x |
> | GEMM | 84 | 463 (CUTLASS) | 0.98-2.05x |
> | Convolution (N=4096) | 131 | 642 (CUDA FlashFFTConv) | 4.7x |
> | Based linear attention | 282 | 89 (Triton) | 3.7-14.5x |
> | Hedgehog linear attention | 316 | 104 (Triton) | 4.0-6.5x |
> | Mamba-2 | 192 | 532 (Triton) | 3.0-3.7x |
> | Rotary | 101 | 119 (Triton) | 1.1-2.3x |
> | Fused layer norm | 146 | 124 (Triton) | 1.0-2.2x |

---

### Meta-Review · Area_Chair_2ZdH · 2024-12-15

**Metareview:**

The paper presents a new framework to simplify writing AI kernels for GPUs while still allowing for high performance. All the reviewers liked the paper for its practical significance. Hence, I recommend acceptance.

**Additional Comments On Reviewer Discussion:**

The rebuttals further demonstrated the practical utility of the proposed method.

---

### Decision · Program_Chairs · 2025-01-22

Accept (Spotlight)